# Recurrent patterns of DNA copy number alterations in tumors reflect metabolic selection pressures

Nicholas A Graham[1,2,3,†] (ID), Aspram Minasyan[1,2,†] (ID), Anastasia Lomova[1,2] (ID), Ashley Cass[1,2], Nikolas G Balanis[1,2], Michael Friedman[1,2], Shawna Chan[1,2] (ID), Sophie Zhao[1,2], Adrian Delgado[1,2], James Go[1,2], Lillie Beck[1,2], Christian Hurtz[4,§] (ID), Carina Ng[4], Rong Qiao[2], Johanna ten Hoeve[1,2], Nicolaos Palaskas[1,2] (ID), Hong Wu[2,5,6], Markus Müschen[4,7], Asha S Multani[8], Elisa Port[9], Steven M Larson[10], Nikolaus Schultz[11,12], Daniel Braas[1,2,13], Heather R Christofk[2,5,13,‡], Ingo K Mellinghoff[12,14,15,16,‡] & Thomas G Graeber[1,2,5,13,17,*] (ID)

## Abstract

Copy number alteration (CNA) profiling of human tumors has revealed recurrent patterns of DNA amplifications and deletions across diverse cancer types. These patterns are suggestive of conserved selection pressures during tumor evolution but cannot be fully explained by known oncogenes and tumor suppressor genes. Using a pan-cancer analysis of CNA data from patient tumors and experimental systems, here we show that principal component analysis-defined CNA signatures are predictive of glycolytic phenotypes, including $^{18}$F-fluorodeoxy-glucose (FDG) avidity of patient tumors, and increased proliferation. The primary CNA signature is enriched for p53 mutations and is associated with glycolysis through coordinate amplification of glycolytic genes and other cancer-linked metabolic enzymes. A pan-cancer and cross-species comparison of CNAs highlighted 26 consistently altered DNA regions, containing 11 enzymes in the glycolysis pathway in addition to known cancer-driving genes. Furthermore, exogenous expression of hexokinase and enolase enzymes in an experimental immortalization system altered the subsequent copy number status of the corresponding endogenous loci, supporting the hypothesis that these metabolic genes act as drivers within the conserved CNA amplification regions. Taken together, these results demonstrate that metabolic stress acts as a selective pressure underlying the recurrent CNAs observed in human tumors, and further cast genomic instability as an enabling event in tumorigenesis and metabolic evolution.

**Keywords** aneuploidy; DNA copy number alterations; genomic instability; glycolysis; metabolism

**Subject Categories** Cancer; Genome-scale & Integrative Biology; Metabolism

**Mol Syst Biol.** (2017) 13: 914

## Introduction

Cancer cells differ from normal cells in that they exhibit aberrant proliferation, resist apoptosis, and invade other tissues (Hanahan & Weinberg, 2011). Modern cancer classification relies on molecular

1   Crump Institute for Molecular Imaging, David Geffen School of Medicine, University of California, Los Angeles, CA, USA
2   Department of Molecular & Medical Pharmacology, David Geffen School of Medicine, University of California, Los Angeles, CA, USA
3   Mork Family Department of Chemical Engineering and Materials Science, University of Southern California, Los Angeles, CA, USA
4   Department of Laboratory Medicine, University of California, San Francisco, CA, USA
5   Jonsson Comprehensive Cancer Center, David Geffen School of Medicine, University of California, Los Angeles, CA, USA
6   School of Life Sciences & Peking-Tsinghua Center for Life Sciences, Peking University, Beijing, China
7   Department of Haematology, University of Cambridge, Cambridge, UK
8   Department of Genetics, M. D. Anderson Cancer Center, The University of Texas, Houston, TX, USA
9   Department of Surgery, Icahn School of Medicine at Mount Sinai, New York, NY, USA
10  Department of Radiology, Memorial Sloan Kettering Cancer Center, New York, NY, USA
11  Marie-Josée and Henry R. Kravis Center for Molecular Oncology, Memorial Sloan Kettering Cancer Center, New York, NY, USA
12  Human Oncology and Pathogenesis Program, Memorial Sloan Kettering Cancer Center, New York, NY, USA
13  UCLA Metabolomics Center, David Geffen School of Medicine, University of California, Los Angeles, CA, USA
14  Department of Neurology, Memorial Sloan Kettering Cancer Center, New York, NY, USA
15  Department of Pharmacology, Weill Cornell Medical College, New York, NY, USA
16  Department of Neurology, Weill Cornell Medical College, New York, NY, USA
17  California NanoSystems Institute, David Geffen School of Medicine, University of California, Los Angeles, CA, USA
    *Corresponding author. Tel: +1 310 206 6122; Fax: +1 310 206 8975; E-mail: tgraeber@mednet.ucla.edu
    †These authors contributed equally to this work
    ‡These authors contributed equally to this work
    §Present address: Division of Hematology and Oncology, Department of Medicine, University of Pennsylvania, Philadelphia, PA, USA

characterization, including examination of genomic DNA mutations and copy number alterations (CNAs; Stuart & Sellers, 2009). Although individual oncogenes and tumor suppressor genes are preferential targets of DNA amplifications and deletions, respectively, the recurrent CNA patterns in tumors cannot be fully explained by canonical cancer genes (Beroukhim *et al*, 2010; Muller *et al*, 2012; Davoli *et al*, 2013). Thus, the unexplained recurrent CNA patterns observed in human cancer subtypes are suggestive of additional, not yet fully defined, selective pressures that are conserved across patients and tumor types (Cahill *et al*, 1999; Sheltzer, 2013; Cai *et al*, 2016). Reports that the cumulative phenotypic effects of many small gene dosage alterations across the genome can impact the resulting tumor copy number landscape (Solimini *et al*, 2012; Davoli *et al*, 2013) illustrate a need to consider more subtle and combinatorial effects to explain the remaining selective forces underlying recurrent CNA patterns observed in human cancers.

One of the fundamental and consequential differences between non-transformed and tumorigenic cells is the reprogramming of cellular metabolism (Hanahan & Weinberg, 2011). The altered metabolism of tumors is thought to benefit transformed cells in several ways. Upregulation of glucose metabolism allows proliferating cells to meet their energy demand through synthesis of adenosine triphosphate (ATP), while increased flux through glycolysis branch pathways provides dividing cells with intermediates necessary for biosynthesis of nucleotides and fatty acids, as well as reducing agents such as glutathione and NADPH (DeBerardinis *et al*, 2008; Cairns *et al*, 2011). Moreover, in addition to glucose, cancer cells frequently upregulate consumption of other metabolites for energy and biomass generation, including glutamine, serine, and glycine (Jain *et al*, 2012; Maddocks *et al*, 2013). Notably, several individual metabolic enzymes have been directly implicated in tumorigenesis (Kim *et al*, 2007; Dang *et al*, 2009; Locasale *et al*, 2011; Possemato *et al*, 2011; Patra *et al*, 2013; Li *et al*, 2014; Wang *et al*, 2014; Xie *et al*, 2014) and/or immortalization (Kondoh *et al*, 2005; Kondoh, 2009; Kaplon *et al*, 2013), suggesting that altered metabolism is not a passive bystander, but rather a driving force of oncogenesis (Yun *et al*, 2009; Zhang *et al*, 2012). Using an integrative analysis of CNA data from human tumors, mouse models of cancer, cancer cell lines, and a murine experimental immortalization system, here we show that the loci of metabolic genes impact the recurrent CNA changes observed in genomically unstable tumors. Our bioinformatic and experimental results support a tumorigenesis model in which copy number changes in metabolic genes contribute to an enhanced glycolytic and proliferative state (see Fig EV1 for a schematic of our overall approach).

# Results

### PCA-defined CNA signatures in human cancers

To develop an unbiased understanding of DNA copy number alterations (CNAs) in cancer, we performed principal component analysis (PCA) of gene-based CNA data derived from comparative genomic hybridization (CGH) microarrays from 15 tumor types available from The Cancer Genome Atlas (TCGA). This pan-cancer PCA revealed a high degree of similarity in CNA profiles between basal breast invasive carcinoma (BRCA basal), lung squamous cell carcinoma (LUSC), ovarian serous cystadenocarcinoma (OV), and serous uterine corpus endometrial carcinoma (UCEC serous) (Fig 1A and Appendix Fig S1). In tumor type-specific PCA, analyzing the four tissue-defined tumor sets of BRCA, LU (lung cancer consisting of LUSC and LUAD [lung adenocarcinoma]), OV, and UCEC revealed two strong PCA-based CNA signatures, termed signatures A and B, in all four cases. Notably, signature A was highly consistent across each tumor type, and reflected the pattern of pan-cancer PC1 loadings (Figs 1B and C, and EV2A–C). In BRCA, signature A tumors were enriched for the basal subtype, p53 point mutations, high numbers of genomic breakpoints, and thus subchromosomal alterations (Figs 1B and D, and EV2D; $P < 0.001$, $P < 0.001$, $P = 2 \times 10^{-4}$, respectively). Signature B BRCA tumors, in contrast, were enriched for luminal type tumors ($P < 0.001$) and exhibited amplifications of the oncogenes *MYC* and *MDM2* and deletion of the tumor suppressor *CDKN2A* (Fig 1B). Amplification of *MDM2* and loss of *CDKN2A* were generally mutually exclusive in signature B tumors (Fig EV2E), reflecting alternate mechanisms for disabling the p53/ARF axis (Sherr & Weber, 2000). In the other tissues, signature A tumors were enriched for lung squamous cell carcinomas, the proliferative subtype of ovarian cancer (The Cancer Genome Atlas Research Network, 2011), and the serous subtype of uterine cancer (Fig EV2A–D). Overall, signature A tumors demonstrated enrichment of p53 mutations, more genomic breakpoints (BRCA, LU, and UCEC), and a higher degree of copy number alterations (LU, OV, and UCEC) than signature B tumors (Appendix Fig S2A and B, *P*-values indicated in figure). In signature A tumors, the per tumor average segment size is on the scale of $1 \times 10^{7}$ base pairs (approximately one-tenth of a chromosome, containing on the order of 100 genes). Overall, the segment sizes in signature A span from focal to arm-length/whole chromosome scale. The per tumor average segment size for signature B BRCA, LU, and UCEC tumors is statistically larger by 1.3-fold to sevenfold (Appendix Fig S2C–E; $P = 2 \times 10^{-4}$ or less). Unlike signature A tumors, the signature B CNA patterns were quite distinct between tumor types, although some commonalities were observed including point mutations in oncogenes such as *KRAS* (LU and UCEC) and amplification of *MYC* (BRCA, LU, and OV). An alternative approach using hierarchical clustering confirmed the existence of the shared pan-cancer CNA signatures across multiple tumor types (signature A), as well as distinct signature subtypes within each of the BRCA, OV, UCEC, and LU tumor types (signature A vs. signature B) (Fig EV3 and Appendix Fig S3, *P*-values of concordance between PCA and clustering approaches indicated in figure). In summary, PCA revealed the CNA signature A as a pattern shared across a subset of tumors in multiple tissue types, as well as several tumor type-specific cases of more distinct signature B patterns (Appendix Fig S1D).

Because altered metabolism is a hallmark of human tumors, we next tested whether the shared CNA signature A was enriched for genes from metabolic pathways. Using CNA-based gene set enrichment analysis over all metabolic pathways defined by KEGG (Kanehisa *et al*, 2014), we found that the conserved profile of core signature A tumors (i.e., OV, BRCA, UCEC, LU) (Fig 1C) was significantly enriched for DNA amplifications of core glycolysis pathway genes (Fig 1E and F, and Table EV1; $P = 0.024$). For example, BRCA

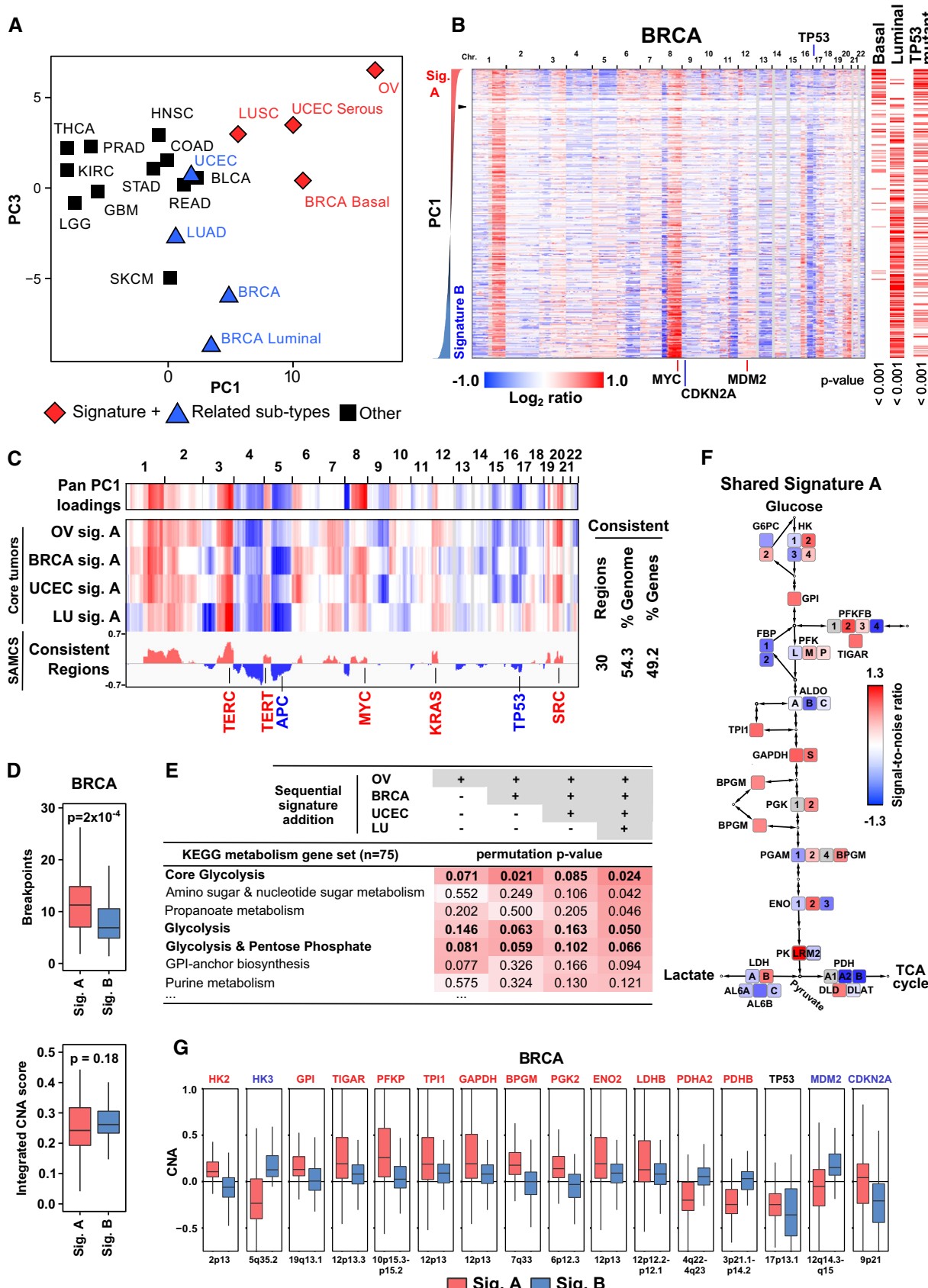

**Figure 1.**

**Figure 1.  Principal component analysis (PCA) reveals a shared CNA signature in breast, lung, ovarian, and uterine carcinomas.**

A   Pan-cancer PCA of copy number data from a balanced, random sampling of tumors of 15 tumor types from The Cancer Genome Atlas (TCGA). The average tumor PC scores for each tumor subtype are shown. Pan-cancer PC1 scores primarily separate diploid from highly aneuploid tumors, while PC2 distinguished GBM from the other tumor types (Appendix Fig S1A and B). Four tumor subtypes with similar PC scores are labeled as "Signature positive (+)". Tumor type abbreviations are as defined by TCGA: ovarian serous cystadenocarcinoma (OV), breast (BR), bladder urothelial (BL), and thyroid (TH) carcinoma (*CA), uterine corpus endometrial carcinomas (UCEC), lung (LU), and head and neck (HN) squamous carcinomas (*SC), skin cutaneous melanoma (SKCM), lung (LU), rectal (RE), colon (CO), stomach (ST), and prostate (PR) adenocarcinomas (*AD), glioblastoma (GBM), low-grade glioma (LGG), and kidney renal clear cell carcinoma (KIRC).

B   Copy number profiles of 873 breast invasive carcinomas (BRCA). Tumors (rows) sorted by tumor-specific principal component 1 (PC1) score with genomic locations listed across the top. PCA identified and distinguished two signatures with similar degrees of variance in the first component. The triangle marks the transition from signature A through diploid samples to signature B. The membership of each tumor with known molecular subtypes (e.g., basal/luminal) and mutant *TP53* status are indicated on the right with red horizontal bars and corresponding permutation-based enrichment *P*-values. Basal and luminal subtype classifications are from a published gene expression-based determination (The Cancer Genome Atlas Research Network, 2012b). Signature A was moderately enriched for claudin-low tumors, whereas the HER2-enriched subtype was not significantly associated with either signature A or B (Fig EV2D). Similar PCA-based distinctions were observed in the LU, OV, and UCEC datasets (Fig EV2A–C).

C   Pan-cancer PC1 loadings and signature A summary profiles for ovarian (OV), breast (BRCA), uterine (UCEC), and lung (LU) tumor types. Summary signatures are normalized gene loci signal-to-noise ratios (SNRs) of the top 10% of PC1-based core signature A tumors compared to normal (non-tumor) samples. Pan-cancer PC1-3 loadings are shown in Appendix Fig S1C. Consistency signatures of conserved amplification and deletions (consistent regions) are indicated by their signed absolute minimum consistency score (SAMCS). The SAMCS is non-zero when all CNA summary signatures have the same sign across all tumor types, and is derived from the absolute value-based minimum summary metric, and then re-signed positive for amplification or negative for deletion (see Materials and Methods). In the bar graph, red and blue denote consistently amplified and deleted regions, respectively. The number or percentage of consistent regions, genome coverage, and gene loci are indicated. The positions of canonical oncogenes, proto-oncogenes, tumor suppressor genes, and telomerase components (*TERC*, *TERT*) present in these regions are indicated.

D   Breast signature A tumors have more DNA breakpoints per chromosome than signature B tumors, but similar levels of copy number alteration (as measured by an integrated CNA score proportional to the extent and absolute value magnitude of amplifications and deletions). *P*-values are indicated (top, Mann–Whitney *U*-test; bottom, Student's *t*-test), and data presented in box (median, first and third quartiles) and whisker (extreme value) plots. The number of breakpoints is inversely proportional to the mean DNA segment length.

E   KEGG metabolism pathway enrichment analysis based on consistent CNA patterns in the core signature A tumors of (A). The OV, BRCA, UCEC, and LU signatures from (C) were sequentially added and only directionally consistent CNA changes were retained (see Materials and Methods). The combined glycolysis–gluconeogenesis (glycolysis) and pentose phosphate pathway was included based on our prior mRNA work identifying the predictive value of this gene set (Palaskas *et al*, 2011). Core glycolysis is a KEGG-defined gene subset (M00001). Table EV1 lists enrichment results for all KEGG metabolism pathways.

F   Schematic showing average signal-to-noise (SNR) metrics of core glycolysis pathway genes plus 6-phosphofructo-2-kinase/fructose-2,6-biphosphatase (PFKFB and TIGAR), lactate dehydrogenase (LDH), and pyruvate dehydrogenase (PDH) in core signature A tumors from (C) and (E). A more detailed listing of the gene names can be found in Materials and Methods.

G   Gene copy number alteration distributions of selected glycolysis genes, *TIGAR*, the tumor suppressor p53, and the p53-associated cancer genes *MDM2* and *CDKN2A* in BRCA tumors. Data are presented in box (median, first and third quartiles) and whisker (extreme value) plots.

Data information: In panels (C, E–G), the top 10% of tumors from each indicated signature based on PC1 scores were used in the analysis. See also Fig EV2 and Appendix Figs S1–S3.

signature A tumors exhibited DNA amplification of most genes from the glycolytic pathway, as well as amplification of lactate dehydrogenase B, deletion of pyruvate dehydrogenase subunits A and B, and amplification of the glycolysis-regulating metabolic enzyme TIGAR (human gene *C12orf5*; Fig 1G). Notably, this shared CNA signature was defined by genome-wide patterns, rather than by single gene loci, which were not consistently altered in all tumors with a strong signature (Appendix Fig S2F). Interestingly, signature A summary profiles from breast, ovarian, uterine, and lung tumors all exhibited hexokinase 2 (HK2) amplification (Fig 1F and G), whereas signature B profiles had primarily either HK3 (BRCA, LU, OV) or HK1 (UCEC) amplification (Fig 1G and Appendix Fig S2G). Thus, PCA identified a shared signature from breast, lung, ovarian, and uterine carcinomas that was enriched for p53 mutations, higher numbers of genomic breakpoints, and CNA of genes from the core glycolysis pathway.

**Elimination of passenger genes via cross-species synteny mapping**

While canonical oncogenes and tumor suppressor genes drive some recurrent DNA copy number alterations, many recurrent CNA regions cannot be fully explained by the presence of known cancer genes (Beroukhim *et al*, 2010; Muller *et al*, 2012; Davoli *et al*, 2013). The conservation of CNA signature A across breast, lung, ovarian, and uterine tumor subsets suggests the existence of one or

more selective pressures that are potentially shared by additional tumor types. In that many of the human signature A conserved regions identified in Fig 1C still span large, chromosome-scale regions, we hypothesized that passenger genes could be diluting the enrichment signal. We subsequently reasoned that a previously reported approach of cross-species comparison of CNA data from human tumors and non-human tumorigenesis models would eliminate passenger genes via synteny mapping (Maser *et al*, 2007; Zhang *et al*, 2013; Tang *et al*, 2014).

For the cross-species analysis, we first empirically determined the full set of human signature A-like tumor tissue types that could be included based on CNA similarity as defined by the pan-cancer PC1 scores from Fig 1A. Namely, we sequentially added each human tumor type in order of decreasing pan-cancer PC1 score and tested whether inclusion improved the overall pathway enrichment. In a parallel analysis, at each sequential step we also added the corresponding tissue-matched mouse epithelial cancer model signature if it existed. Mouse CNA signatures obtained from the literature and public repositories included signatures for genetically engineered mouse models of mammary (breast) cancer (Brca) (Drost *et al*, 2011; Herschkowitz *et al*, 2012), melanoma (Skcm) (Viros *et al*, 2014), glioblastoma/high-grade astrocytoma (Gbm) (Chow *et al*, 2011), and prostate cancer (Prad) (Ding *et al*, 2012; Wanjala *et al*, 2015), as well as *in vitro* mouse epithelial cell models of bladder (Blca), colorectal (Coad), and kidney (Kirc) cancer (Padilla-Nash *et al*, 2013).

Using human tumors only, the sequential enrichment analysis showed a general improvement through the addition of the COAD tumor type (Fig 2A, dotted orange line; average enrichment signal of the top 10 metabolic pathways). Moreover, sequential inclusion of the corresponding tissue-matched mouse cancer model CNA signatures substantially enhanced the overall pathway enrichment results with the peak occurring at HNSC, a sequentially adjacent tumor type that has similar pan-cancer PC1 score with COAD (Fig 2A, solid blue line and arrow indicating the peak). Overall pathway enrichment then decreased as additional, less copy number similar (lower pan-cancer PC1 scores) tumor types were added.

The improvement upon including the mouse signatures supports the hypothesis that the cross-species analysis eliminates passenger genes and reveals pathways whose genes are enriched in the resulting cross-species consistent CNA regions. Taken together, this pan-cancer and cross-species analysis demonstrates that a combination of nine human tumor types (OV, BRCA, UCEC, LU, BLCA, READ, SKCM, COAD, and HNSC) and four corresponding mouse tumor models (Brca, Blca, Skcm, and Coad) gives the strongest overall enrichment signal strength across all metabolic pathways (Fig 2A).

Examining the highly ranked individual pathways from this optimized tumor type combination, we found that the carbohydrate metabolism pathway "amino sugar and nucleotide sugar metabolism" (hsa00520) was ranked first and glycolysis–gluconeogenesis (hsa00010, henceforth called glycolysis) was ranked second among KEGG metabolism gene sets (Fig 2B and Table EV1). As in the average of top 10 pathways (Fig 2A), the enrichment score for the glycolysis gene set improved as more human tumor and mouse models were sequentially included up through the additions of HNSC and Coad (Fig 2C). Compared to our previous enrichment analysis of the four core signature A tumors (Fig 1E), the permutation *P*-value of the glycolysis gene set improved from 0.05 to 0.001, reflecting that our pan-cancer and cross-species analysis has eliminated passenger genes in the non-consistent CNA regions. Thus, we hereafter refer to the consistent signature pattern from OV up through and including HNSC and Coad as "expanded signature A". Of particular note, upon expanding our analysis to include 1,321 gene sets from the MSigDB Canonical Pathways (CP) database, glycolysis remained a top result as the third ranked pathway, with the "glutathione-conjugation" pathway (involved in cell detoxification and oxidative stress responses) also scoring strongly (Fig 2B and Table EV2).

To visualize the pan-cancer and cross-species conserved genomic regions, we plotted the consistency profiles of the expanded signature A human and mouse tumors (Fig 2D). Examination of conserved regions revealed that *HK2, TPI1, GAPDH, PGAM2, ENO2,* and *LDHB* glycolysis genes contribute to the cross-species consistency signal. Among canonical oncogenes, *MYC* and *KRAS* were also present in the amplification regions of the expanded signature A human tumors and mouse models. Due to the well-documented and clinically relevant role of glycolysis and pentose phosphate pathways in tumorigenesis, we chose to further examine the recurrent amplification of the set of these gene loci in subsequent analyses. Equally important for functional validation of these candidate pathways, the activity of glycolysis can be directly and indirectly measured by many assays in both patients (e.g., FDG-PET imaging) and experimental systems.

## CNA signatures are predictive of glycolysis

In that the CNA-consistent region-defined signature A is enriched for core glycolysis genes, we next tested whether signature A patient tumors *in vivo* were associated with increased tumor glycolysis. To assign a signature A score to a set of FDG-PET-imaged breast cancers (Palaskas *et al*, 2011), we projected CNA data from these tumors onto a PCA of the four core human tumor tissue types (OV, BRCA, UCEC, and LU, Fig 1C). We found a strong correlation between the strength of CNA signature A and the measured FDG-PET standardized uptake values (SUVs) (Fig 3A and B; Pearson rho = 0.94, $P = 5 \times 10^{-5}$). A similar analysis using all nine expanded signature A tumor types demonstrated equivalent results (Appendix Fig S4A; Pearson rho = 0.92, $P = 2 \times 10^{-4}$). Thus, the CNA-defined signature A is associated with increased FDG uptake in human primary tumors *in vivo*.

We next asked whether the CNA signature A-defined tumors associated with high FDG uptake also had RNA-based signatures of increased glycolysis. First, we compared signature A tumors to signature B tumors using RNA-based enrichment analysis (GSEA). We analyzed both BRCA and LU tumors because these tumor types show distinct signature A and signature B subtypes. (OV signature B is highly similar to the pan-cancer signature A pattern, and thus does not provide a differential test (Fig EV2B). UCEC tumors were not included due to a lack of sufficient paired RNA and DNA profiling data.) In the enrichment analysis, we included a gene set consisting of genes from the glycolysis and pentose phosphate pathways that were upregulated in FDG-high BRCA tumors, as defined by our previous work (Palaskas *et al*, 2011). In the GSEA, the empirically defined FDG-high gene set ranked number one overall (NES = 2.5, permutation *P*-value = $2 \times 10^{-4}$), confirming that Sig A tumors have glycolysis RNA expression profiles matching those of FDG-high tumors (Fig EV4A). Furthermore, BRCA and LU signature A tumors were significantly enriched for genes from the full glycolysis and pentose phosphate pathways (overall rank 4th of 76 pathways, NES = 2.0, permutation *P*-value = $6.8 \times 10^{-3}$), as well as other glycolysis-related pathways (Table EV3). This analysis of RNA expression data is consistent with the enrichment of glycolysis-associated pathways at the DNA copy number level (Fig 1E and 2B), and points to glycolysis-related pathways as selection targets for upregulation in signature A tumors.

To further explore the RNA expression data, we predicted the glycolytic phenotypes of the core signature A tumor types (BRCA, LU, and OV) using RNA-based weighted gene voting (WGV) (Golub *et al*, 1999) and our previously defined FDG prediction model (Palaskas *et al*, 2011). UCEC tumors were again excluded because there were not a sufficient number of samples with paired RNA and DNA data. We found that RNA-based predictions of high glycolysis were associated with the signature A end of tumor type-specific PC1 for BRCA and LU (Fig EV4B–D). Signature A tumors were predicted to be significantly more glycolytic than signature B tumors for BRCA and LU (*P*-values of $2.4 \times 10^{-17}$ and $5.7 \times 10^{-18}$, respectively, Fig EV4E). There was no significant differential trend in OV tumors, potentially because almost all of these tumors are genomically unstable (integrated CNA scores > 0.2) and the signature B of OV is relatively signature A-like (Fig EV2B and Appendix Fig S1D). The OV predictions were consistent with predicted high glycolysis across all tumors (Fig EV4E). To control

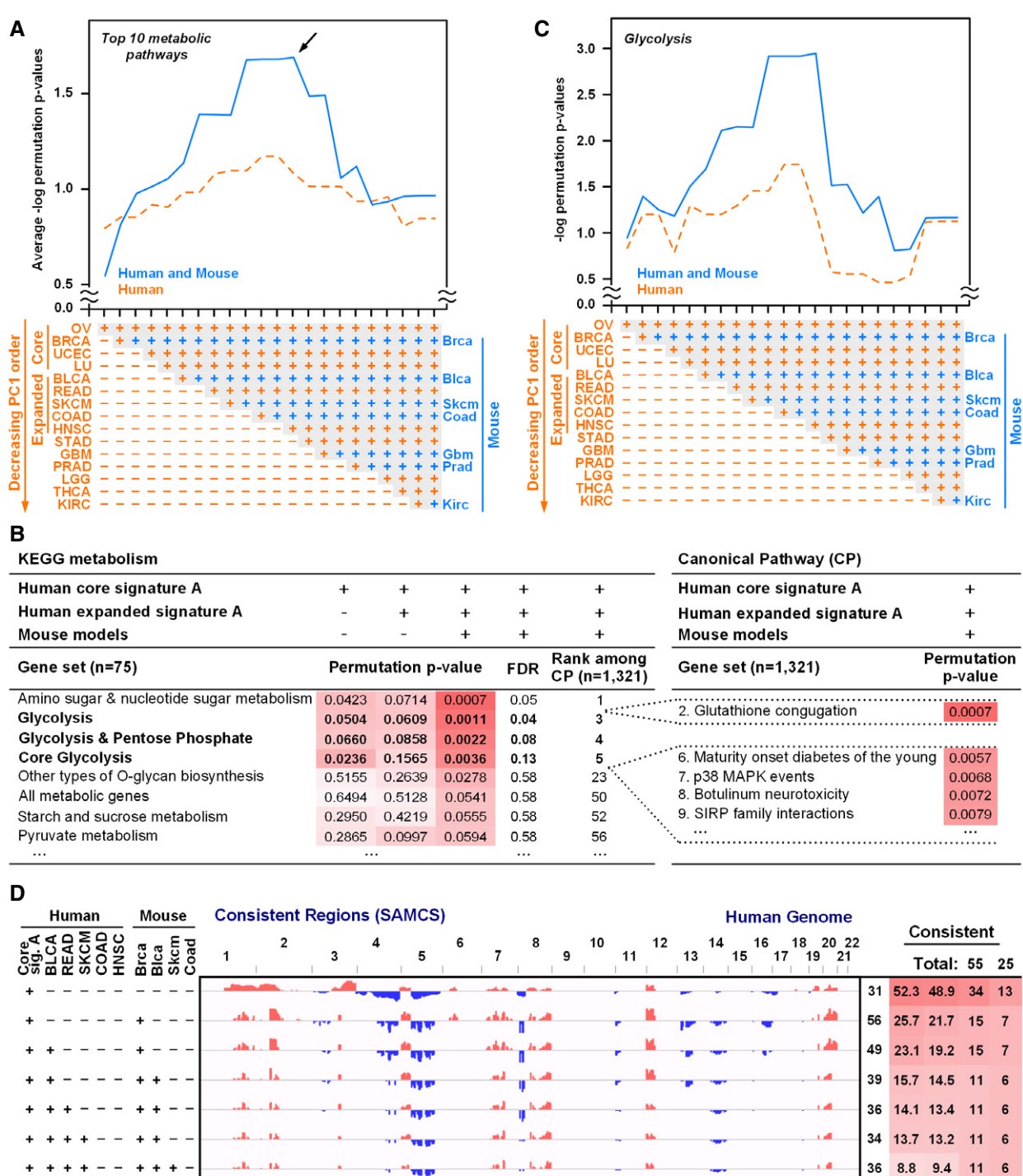

**Figure 2.**

**Figure 2.  Pan-cancer and cross-species analysis reveals enrichment of glycolytic genes in conserved amplification regions.**

A    Average pathway enrichment signal strength for the top 10 enriched KEGG metabolism pathways based on the consistent CNA patterns across multiple tumor types. The four core CNA signature A patterns are defined as in Fig 1C. CNA signatures for other tissue tumor types are defined by the signal-to-noise ratios (SNRs) of all tumors compared to tissue-matched normal (non-tumor) samples. Starting with the four core signature A tumor types from Fig 1 (OV, BRCA, UCEC, and LU), additional human (orange) or human and mouse (blue) signatures were sequentially added based on their decreasing tissue type pan-cancer PC1 score from Fig 1A (see Materials and Methods). The resulting sequentially restricted consistency signatures are illustrated in (D). A maximal peak (indicated by an arrow) in the enrichment signal of this sequential analysis is seen after adding tumors through human HNSC and mouse Coad (no mouse Hnsc signature was available). Enrichment signal strength is based on the negative log frequency of equivalent enrichment by chance (i.e., permutation *P*-value). Mouse tumor type abbreviations are shown with lower case letters but otherwise match the human TCGA abbreviations.

B    Enrichment analysis of metabolism (75 KEGG pathways) and canonical pathways (CP; 1,321 MSigDB canonical pathways). Permutation *P*-values for the consistent CNA patterns from the four human core signature A tumors (defined in Fig 1C), the human expanded signature A (from A), and the human expanded signature A combined with corresponding mouse cancer models (from A). FDR calculations are described in Materials and Methods. Tables EV1 and EV2 list all sequential enrichment results for KEGG metabolism and canonical pathways, respectively.

C    Enrichment score for the glycolysis pathway upon sequential addition of human tumors and mouse models (as in A) showing a maximal peak in the enrichment signal corresponding to the expanded HNSC–Coad signature.

D    Genome view of the sequential consistency signatures used in the enrichment analysis of (A–C). The locations of consistently amplified glycolysis genes, canonical oncogenes, and tumor suppressors are indicated. Genes listed in parentheses (i.e., *RPIA*, *TERC*, *TERT*, *APC*, and *TP53*) were consistently altered across the human tumors but not consistently altered in the mouse models, due in part to the inclusion of p53 genetic knockout mouse models (*TP53*) and known differences in human and mouse telomere maintenance (*TERC*, *TERT*) (Sherr & DePinho, 2000). The synteny graph at the bottom indicates the syntenic mouse chromosome number and thus the broken synteny regions between human and mouse genomes. This genome view chronicles how the sequential pan-cancer and cross-species analysis of conserved amplification and deletion loci greatly reduces the percentage of consistent copy number alterations, thereby reducing the percentage of the genome implicated as candidate driver regions.

for the general increase in glycolysis predictions at increased levels of genomic instability, we also analyzed the correlation between glycolysis predictions and PCA scores for several range windows of integrated CNA scores. This analysis demonstrated that RNA-based predictions of glycolysis were significantly correlated with PC1 scores in BRCA and LU tumors with high integrated CNA scores (permutation *P*-value for BRCA and LU < $1 \times 10^{-6}$ each, Fig EV4F). Taken together, the DNA-defined signature A tumors exhibit RNA expression patterns consistent with increased FDG uptake. These RNA-based results support the more downstream activity-based findings of elevated FDG uptake in signature A tumors (Fig 3A and B).

Returning to the result of the CNA-defined signature A being associated with increased FDG uptake in human primary tumors *in vivo*, we next asked which sets of metabolic gene loci copy number levels were most predictive of glycolytic phenotypes. We performed a CNA-based weighted gene voting (WGV) analysis to predict the glycolytic phenotypes of breast tumors and breast cancer cell lines (Neve *et al*, 2006) using individual gene sets from the KEGG metabolic pathways database (Kanehisa *et al*, 2014). Gene weights were calculated from each of our four tumor type CNA "training" signatures. Specifically, we tested the ability of individual metabolic pathways to predict (i) FDG uptake in patient primary breast tumors and (ii) the lactate secretion of a panel of 32 breast cancer cell lines (Hong *et al*, 2016). Averaging results across these two test cases revealed that genes from the glycolysis and pentose phosphate pathway were most predictive of these metabolic phenotypes (Fig 3C and Table EV4; *P* = 0.01). Moreover, signature A-based predictions were predictive of lactate secretion for basal cell lines more so than for luminal lines (Fig 3D and E, and Appendix Fig S4B), consistent with the observed basal-type tumor enrichment in signature A samples (Figs 1B and EV2D). Thus, consistent with the gene expression-based predictions above, the glycolysis and pentose phosphate pathway DNA copy number alterations from signature A are predictive of glycolytic phenotypes of primary human breast tumors and cancer cell lines.

**Experimental recapitulation of tumor CNA signatures**

Having demonstrated that the glycolytic pathway is statistically associated with genome-wide DNA copy number patterns, we sought an experimentally tractable system that would allow us to test the hypothesis that recurrent patterns of DNA amplification reflect metabolic selection pressures. We thus derived a panel of immortalized mouse embryonic fibroblasts (MEFs) using the classical 3T9 protocol (Todaro & Green, 1963). In this experimental system, under standard culture conditions (e.g., 21% $O_2$), diploid cells undergo a crisis-associated event that increases tolerance for genomic instability and allows them to escape senescence and evolve into cells with chromosomal instability and genomic aberrations (Fig 4A and Appendix Fig S5A; Sun & Taneja, 2007). This system has been used to study core cancer phenotypes such as proliferation, anti-apoptosis, and chromosomal instability (Lowe *et al*, 1993; Gupta *et al*, 2007; He *et al*, 2007; Sotillo *et al*, 2007; Sun *et al*, 2007; Weaver *et al*, 2007) and is one of the few experimentally tractable cancer models involving spontaneous genomic instability (Sherr & DePinho, 2000). In addition, because this system is not driven by strong oncogenes (e.g., KRAS mutation), it allows for complex CNA signatures to evolve from a combination of individual, presumably weaker, DNA alteration events.

We profiled the genome-wide copy number of 42 independent MEF sublines by CGH microarray (Fig 4A). Most samples were profiled after immortalization (post-senescence), with a few profiled before or during senescence. Analysis of this CNA data by PCA revealed that the MEF system recapitulated a two-signature pattern (signatures A and B). These two signatures were generally orthogonal, with the exception of a few "mixed" samples that had CNA characteristics of both signatures A and B (Appendix Fig S5B). As anticipated from prior MEF studies, immortalized MEF cells exhibited an increased number of genomic breakpoints and a higher degree of copy number alterations than the diploid, pre-senescent MEF cells (Appendix Fig S5C and D). Importantly, the MEF-derived signature A resembled the shared signature A pattern derived from

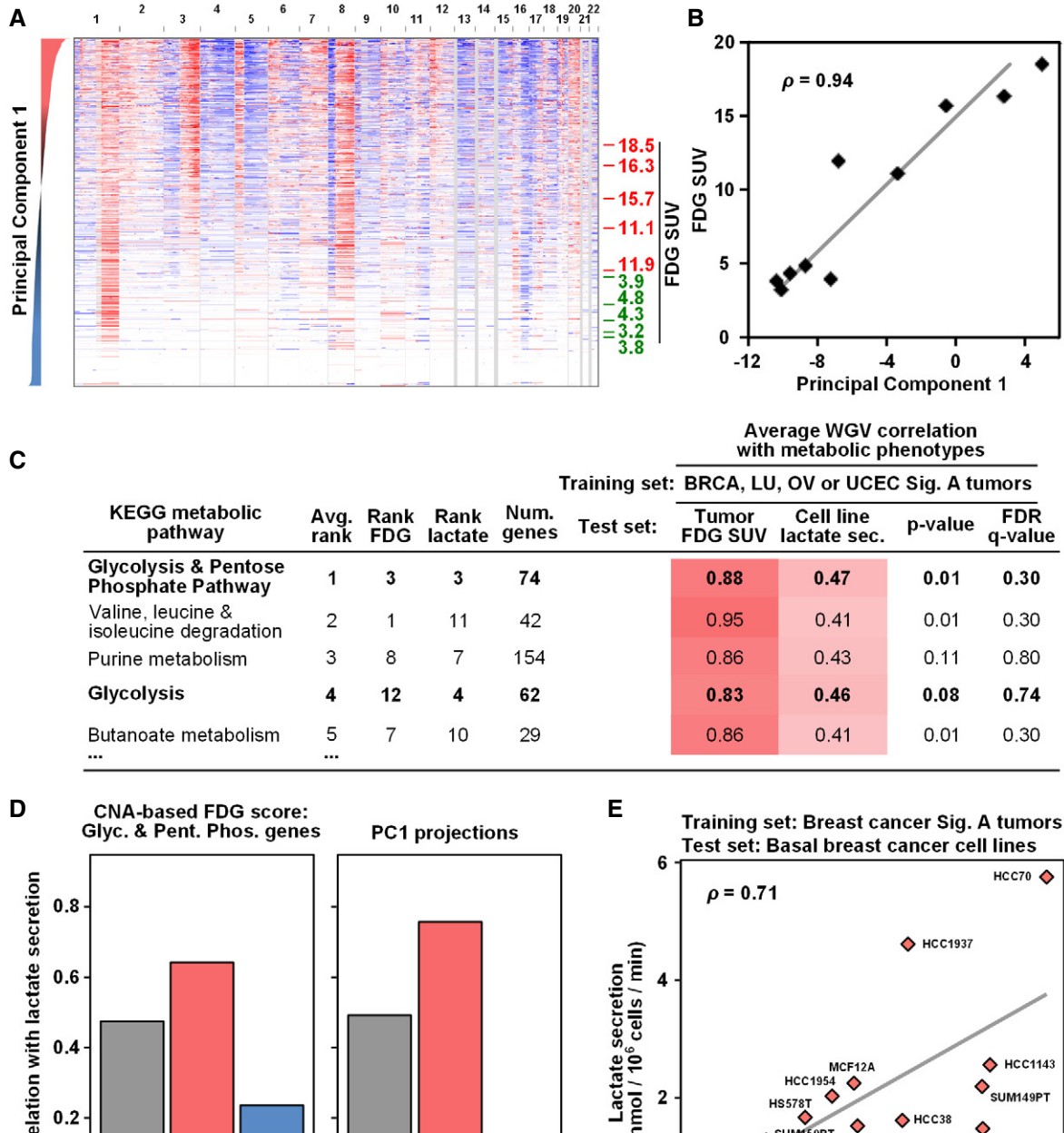

Figure 3.

human breast, lung, ovarian, and uterine tumors. In particular, the MEF-derived signature A was characterized by *p53* mutation and chromosome loss at the p53 locus, more genomic breakpoints, subchromosomal-sized alterations, and a higher degree of copy number alteration (Fig 4A and B, and Appendix Table S1). Additionally, this experimental system allowed us to profile the same MEF lines at subsequent passages, and a paired statistical analysis of five

evolving signature A lines revealed genomic regions changing from mid- to late passage (Fig 4A and Appendix Fig S6A and B). Similar to the shared human tumor signature A, both the MEF signature A and the evolving MEF signature demonstrated DNA amplifications of glycolysis and glycolysis-related genes (Table EV2). In particular, both human and mouse CNA signatures A included amplification of *Hk2, Bpgm, Rpia, Tigar* (mouse gene *9630033F20Rik*), *Eno2, Tpi1,*

**Figure 3.  PCA-based CNA signatures are predictive of breast cancer glycolytic metabolism *in vivo* and breast cancer cell line metabolism *in vitro*.**

A    PC1-sorted copy number profiles of a balanced, random sampling of tumors from the four core CNA-consistent tumor types (breast, lung squamous, ovarian, and uterine carcinomas, Fig 1A) along with copy number profiles from primary breast carcinoma tumors with glycolytic levels imaged *in vivo* by [18]F-fluorodeoxy-glucose positron emission tomography (FDG-PET) (Palaskas *et al*, 2011). On the right, red and green values indicate high and low FDG standardized uptake values (SUVs), respectively, for FDG-imaged tumors.

B    FDG uptake values in FDG-PET measured tumors are highly correlated with CNA signature A PC1 score ($\rho$, Pearson rho correlation; *P*-value = $5.3 \times 10^{-5}$).

C    CNA values for genes from glycolysis and pentose phosphate pathways have stronger predictive power of FDG-PET SUV in breast tumors and of lactate secretion (sec.) in breast cancer cell lines than other metabolism pathway-based sets of genes. The table indicates the correlation between weighted gene voting (WGV)-based predictions on the test sets and the measured metabolic phenotypes. WGV was performed with individual training on signature A tumors from each of the core four tumor types (breast (BRCA), lung (LU), ovarian (OV), and uterine (UCEC); top 10% signature A compared to normal (non-tumor) samples), the voting predictions were averaged, and compared to the measured metabolic phenotype. *P*-values were assessed by permutation analysis. See Table EV4 for all pathways.

D, E    Signature A-based WGV predictions and signature A-based PC1 projections (as in B) have stronger correlation with measured lactate secretion for basal-like breast cancer cell lines than for luminal-like lines (D). See Appendix Fig S4B for luminal cell line data. Basal and luminal subtype classifications are from a published gene expression-based determination (Neve *et al*, 2006). The breast cancer-based glycolysis and pentose phosphate pathway (G & PP) WGV predictions are shown as a representative case (E).

Data information: See also Appendix Fig S4B.

*Gapdh*, *Ldhb*, *Kras* (mouse chr. 6), and *Pgk2* (mouse chr. 17) (Figs 1G and 4A, and Appendix Fig S6C).

In contrast, the MEF-derived signature B was characterized by amplification of *Mdm2* or deletion of *Cdkn2a* (Ink4a/Arf) and fewer overall copy number alterations. *Mdm2* amplification and *Cdkn2a* loss are alternative mechanisms for inactivating p53 function in human tumors and in MEFs (Sherr & DePinho, 2000), but as found here result in a distinct CNA signature as compared to the p53 mutation-associated signature A. As in human BRCA (Fig EV2E), amplification of *Mdm2* and loss of *Cdkn2a* were generally mutually exclusive in signature B MEFs (Appendix Fig S6D), reflecting alternate mechanisms for disabling the p53/ARF axis (Sherr & Weber, 2000). Additionally, *Mdm2* amplification tends to co-occur with *Hk1* amplification, both loci being located on mouse chr. 10. Thus, the signature B cases are associated with an alternate HK amplification, similar to our finding in signature B human tumors (HK1 or HK3 rather than HK2) (Fig 1G and Appendix Figs S2G and S6C). Immunoblotting confirmed that signature A MEF and signature B MEF lines generally had increased expression of Hk2 and Hk1 protein, respectively (Appendix Fig S6E and F).

To further characterize the association between p53 loss and CNA signatures, we profiled the CNA patterns of 29 independent p53$^{-/-}$ MEF sublines derived in standard 3T9 culture conditions. Comparison of the p53$^{-/-}$ MEF CNA patterns to wild-type MEFs by PCA demonstrated that the CNA patterns of p53$^{-/-}$ MEFs resemble the wild-type MEF signature A pattern, with no signature B-like sublines observed (Appendix Fig S7). p53$^{-/-}$ MEFs do not undergo senescence (Olive *et al*, 2004), and consistent with this, we observed that they tended to have less strong copy number alterations. In summary, strong p53 functional loss (p53 mutation or genetic loss) tends to lead to the CNA signature A pattern, which is associated with a higher degree of copy number alterations (higher integrated CNA score and more breakpoints) and Hk2 amplification, while weaker or less complete p53 functional loss (e.g., mediated by *Mdm2* amplification or *Cdkn2a* loss) is associated with an alternative signature (signature B).

## Numerical and structural chromosomal abnormalities

Next we sought to understand how CNA signatures revealed by CGH relate to numerical and structural aneuploidy (i.e., whole chromosomal and subchromosomal gains or losses, respectively). Using

propidium iodide-based DNA staining, we found that immortalized MEF cells had increased total DNA content compared to pre-senescent MEFs (Appendix Fig S8A–C). Similar results were observed in a comparison between a genome-stable, close to diploid, immortalized human mammary epithelial cell line (MCF10A) and a human breast cancer cell line with a high degree of DNA copy number alterations (Hs578T). In addition, we used spectral karyotyping (SKY) to assess the numerical and structural chromosomal aberrations in immortalized MEF cells. Profiling a representative signature A MEF subline just after immortalization (subline H1; passage 25, P25), SKY revealed increased total DNA content (105 ± 36 chromosomes) with nearly all chromosomes having experienced whole chromosome gains (Appendix Fig S8D, E and H). In addition, there was substantial cell-to-cell heterogeneity in the number and type of chromosomes at passage 25. Profiling the same MEF cell line 23 passages later (P48), SKY revealed that the average number of chromosomes had slightly decreased and stabilized (84 ± 12 chr), as has been reported previously (Hao & Greider, 2004), and clonal markers had begun to emerge (e.g., translocation (3:16)) (Fig 4C and Appendix Fig S8F, G and I). We also observed a substantial number of double minutes in some of the cells examined at P48. As expected, the most strongly amplified chromosomes by SKY were scored as gains in CGH data (e.g., chrs. 3 and 6 in subline H1 at P48/49). Since array CGH analysis is normalized by input DNA quantity, genomic regions scoring as negative on a log$_2$ CNA plot can be greater than diploid if most other chromosomes have amplified to an even greater extent (e.g., trisomy chrs. 1, 2, and 7 in subline H1 at P48/49). Similar results have been reported for karyotyping and CGH results of human cancer cell lines (Kytölä *et al*, 2000). Taken together, these data reveal that immortalized MEFs experience substantial numerical and structural chromosomal abnormalities, similar to what is observed in human tumors, and further support that selection for optimized rearranged genomes occurs.

## Senescence-associated oxidative stress as a selective force for copy number alterations

The MEF system allowed us to investigate the selective pressures driving copy number changes during immortalization. Because MEFs cultured under physiological oxygen conditions (3% oxygen) undergo little to no senescence and exhibit less DNA damage

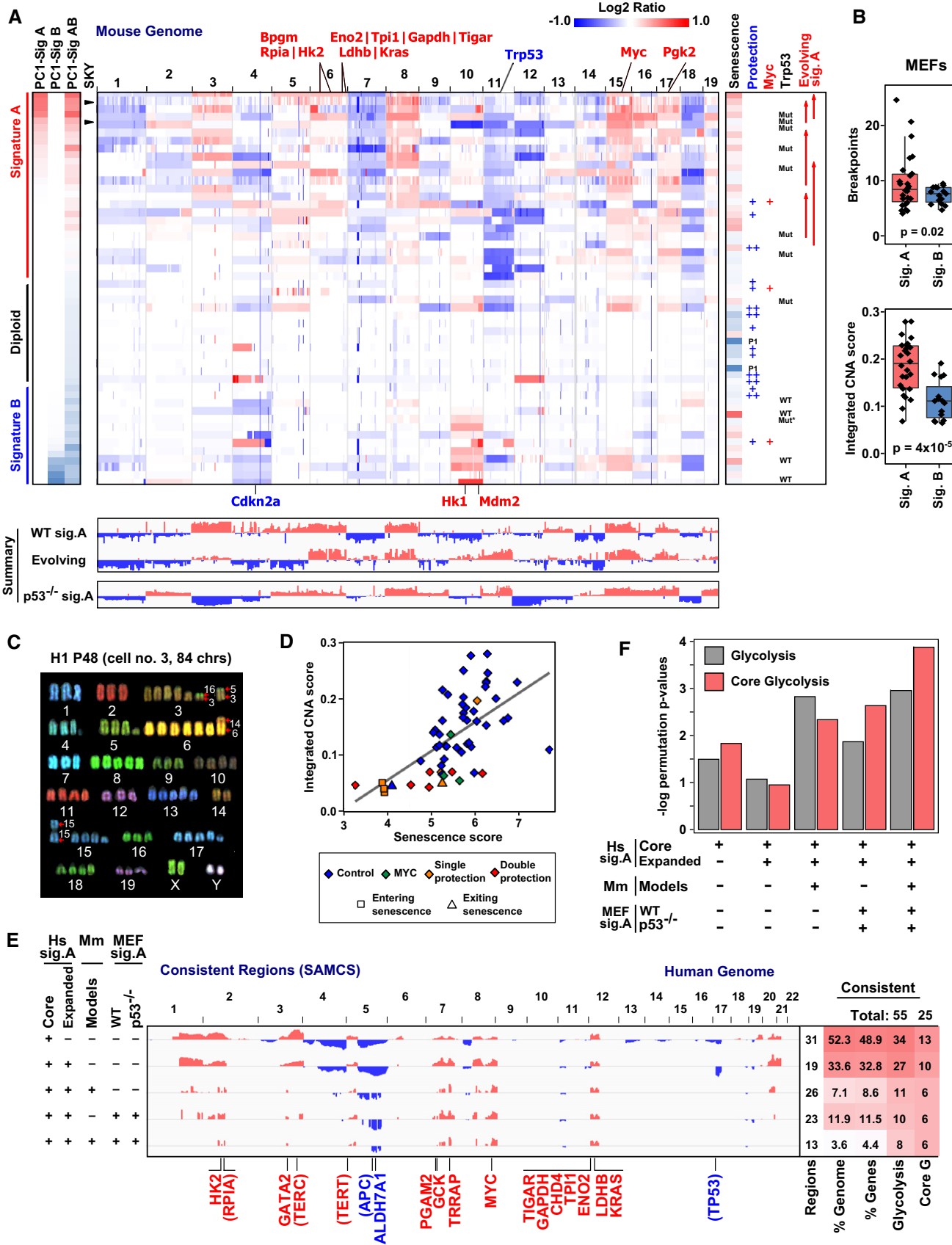

Figure 4.

**Figure 4.  A mouse embryonic fibroblast (MEF) immortalization system recapitulates the two-signature CNA patterns and glycolysis gene CNA enrichment observed in human tumors and mouse models.**

A   Top: Copy number profiles of 59 samples from 42 independent mouse embryonic fibroblasts (MEF) sublines before, during, and after senescence recapitulate the two-signature CNA patterns observed in human tumors. Twenty-two sublines evolved to Sig. A patterns, 13 sublines exhibited Sig. B, and seven sublines remained diploid-like due to redox protection. PC1 scores for analysis of only signature A MEFs (PC1-Sig A), only signature B MEFs (PC1-Sig B), or all MEFs (PC1-Sig AB) are indicated on the left. Also indicated are a metric of the degree of senescence observed during immortalization (senescence score), protection by 3% $O_2$-based hypoxia or catalase (+), or both (++) during immortalization, CNA profiling at passage 1 (P1), exogenous MYC expression (which enabled cells to bypass senescence), and *Trp53* sequencing status (* indicates a non-severe mutation p.183D > E, blank indicates not sequenced, additional p53 sequencing information in Appendix Table S1). Signature A MEF lines profiled at more than one passage number are indicated by the start (earlier passage) and end points (later passage) of the upward "evolving signature A" arrows. For comparison to (C), the early (25) and late (49) passages of the H1 subline that were characterized by spectral karyotyping (SKY) are indicated on the left by triangles. The indicated chromosome 6 region includes loci for *Bpgm*, *Hk2*, *Rpia*, *Eno2*, *Tpi1*, *Gapdh*, *Tigar*, *Ldhb*, and *Kras*. Bottom: Summary signatures of amplification and deletion loci in copy number profiles (normalized PC1 loadings) are shown for MEF WT and p53$^{-/-}$ signature A (Sig. A) samples. A paired *t*-test analysis of signature A MEF lines profiled at more than one passage number revealed genomic regions associated with mid- to late-passage CNA evolution (log$_{10}$ *t*-test *P*-value signed positive for amplifications, negative for deletions; labeled "Evolving"). Individual profiles of p53$^{-/-}$ signature A samples are not shown.

B   Signature A MEFs have more DNA breakpoints and a larger degree of copy number alteration (integrated CNA score) than signature B MEFs. Signature A MEFs had a subset of cases with large numbers of breakpoints per chromosome (> 10) that were not observed in the signature B MEFs (hypergeometric *P*-value = 0.02, Student's *t*-test *P*-value = $4 \times 10^{-5}$). Data are presented in box (median, first and third quartiles) and whisker (extreme value) plots.

C   Spectral karyotyping (SKY) of the H1 MEF subline at passage 48. Whole chromosomal gains of varying extent are observed for all chromosomes with some chromosomes experiencing translocation, for example, t(3:16), and deletion, for example, del(4). Additional chromosome spreads and a summary table for P25 and P48 karyotypes are shown in Appendix Fig S8. CGH profiles for H1 cells are in (A) (indicated by the triangles in the SKY column) and Appendix Fig S6A and B.

D   The amount of senescence demonstrated during MEF line derivation (senescence score, see Materials and Methods) is highly correlated with the degree of copy number alterations obtained (integrated CNA score) (Spearman rho = 0.66, *P*-value = $3 \times 10^{-8}$). Single protection indicates protection by 3% $O_2$ culture conditions or media supplementation with 250 U/ml catalase. Double protection indicates that cells were cultured at 3% $O_2$ with catalase.

E   Conserved amplification and deletion loci in copy number profiles across core and expanded signature A human tumor types (Hs Sig. A), corresponding mouse tumorigenesis model signatures (Mm models), and the MEF immortalization system (MEF Sig. A). The human and mouse signatures are as defined in Fig 2, and the wild-type MEF signatures are from (A). In the human consistent regions, the 52.3% consistently altered gene loci include *bona fide* tumor suppressor genes (p53), oncogenes (MYC, KRAS), telomere components (TERC, TERT), and core glycolysis genes (e.g., HK2). As in Fig 2D, genes listed in parentheses were consistently altered across the human tumors but not consistently altered in the mouse. The tissue-specific, non-lymphoid oncogenes from the COSMIC database that are in a conserved amplification or deletion region (TRRAP, CHD4) are listed for completeness (Forbes *et al*, 2015).

F   Enrichment signal strength (negative of log permutation *P*-value) for the glycolysis and core glycolysis gene sets improve when MEF CNA signatures are added to the human and mouse CNA signatures from Fig 2. The degree of increased enrichment upon addition of the MEF signatures is quantitatively similar to adding mouse tumorigenesis model signatures. Adding all signatures into the analysis further strengthens the enrichment (last column). Table EV2 lists results for the enrichment analysis of the remaining MSigDB canonical pathways when human core and expanded signature A, mouse models, and MEF immortalization models are sequentially combined.

Data information: See also Appendix Figs S5–S9.

(Parrinello *et al*, 2003), we tested whether oxidative stress-induced senescence is a selective pressure for copy number alterations. To protect cells from oxidative stress, we cultured MEFs in 3% oxygen in media supplemented with or without catalase, an enzymatic scavenger of reactive oxygen species (ROS) (Halliwell, 2003; Graham *et al*, 2012). When doubly protected from oxidative stress, MEFs did not undergo senescence and maintained relatively diploid genomes (Fig 4A and Appendix Fig S5A). In addition, we observed a variation in the degree of senescence experienced by MEFs derived in atmospheric oxygen concentrations (21%) (Appendix Fig S5A). Upon calculating the degree of senescence encountered by each subline (senescence metric described in Materials and Methods), we found a correlation between senescence and the degree of copy number alterations (Fig 4D, Spearman rho = 0.66, $P = 3 \times 10^{-8}$; and Appendix Fig S9A). Furthermore, p53$^{-/-}$ cells did not undergo senescence (Olive *et al*, 2004) and exhibited less strong copy number alterations than wild-type signature A MEFs (Appendix Fig S7B; $P = 9 \times 10^{-4}$). Taken together, our results implicate senescence-associated redox stress as one of the selective forces driving the copy number alterations recurrently observed in human tumors.

**MEF CNA signatures recapitulate recurrent patterns from human tumors and further implicate the glycolysis pathway in shaping the cancer genome**

Our pan-cancer and cross-species analysis revealed that the glycolysis pathway is highly enriched in conserved CNA regions (Fig 2).

Because the MEF signature A is qualitatively similar to the human expanded signature A, we next asked how inclusion of our MEF signatures would affect metabolic and canonical pathway enrichment analysis. Although the MEF immortalization system utilizes fibroblast cells and the mouse tumors are epithelial in origin, both types of models share a similar cross-species consistency with the expanded signature A tumor types (Appendix Fig S9B). When the human tumors signatures were combined with the MEF signatures, consistent genome regions were reduced to 3.6% of the genome spread over 13 conserved regions (Fig 4E). In cross-species pathway enrichment analysis, MEF CNA signatures added a similar amount of enrichment signal to human tumor signatures as do the non-MEF mouse model signatures, and when combined together, the enrichment was even stronger (e.g., glycolysis and core glycolysis pathways, Fig 4F). When all signatures are used, glycolysis was the top-ranked enriched pathway out of 1,321 MSigDB canonical pathways (Table EV2). Thus, including the MEF signatures in the cross-species analysis further implicates the glycolysis pathway in shaping the cancer genome.

Examination of the cross-species genomic regions conserved in human tumors, mouse models of cancer, and the MEF immortalization system revealed consistent amplification of five regions containing the COSMIC database-enumerated oncogenes *GATA2* (human chr. 3; Fig 4E, last SAMCS line), *TRRAP* (chr. 7, region 4), *MYC* (chr. 8, region 1), *CHD4* (chr. 12, region 1), and *KRAS* (chr. 12, region 2) (Forbes *et al*, 2015). Notably, four of the cross-species conserved CNA genomic regions included genes from the glycolysis

pathway: *HK2* (chr. 2), *GCK* and *PGAM2* (chr. 7, region 1), *ENO2, TPI1,* and *GAPDH* (chr. 12, region 1, which includes *TIGAR*), and *LDHB* (chr. 12, region 2, which also includes *KRAS*). In both human tumors and the MEF immortalization system, the genomic region harboring *TIGAR–GAPDH–TPI1–ENO2* was separated from the *LDHB–KRAS* region by a deletion-prone region that includes the tumor suppressor *CDKN1B* (Appendix Fig S9C–E).

To investigate whether amplification of the DNA cross-species conserved CNA regions results in upregulated RNA expression levels, we examined the correlation between DNA copy number and RNA expression levels in BRCA, LU, and OV tumors for glycolytic genes and, as a point of comparison, known oncogenes (Appendix Fig S10). We examined the average correlation across BRCA, LU, and OV signature A tumors for all genes in the 12 cross-species conserved regions. We found that three glycolytic genes (*LDHB, TPI1,* and *GAPDH*) and one oncogene (*KRAS*) exhibited strong DNA–RNA correlation ($r > 0.66$), three glycolytic genes (*HK2, ENO2,* and *PGAM2*) and three oncogenes (*MYC, CHD4,* and *TRRAP*) exhibited moderate correlation ($0.2 < r < 0.5$), and only one glycolytic gene (*GCK*) and one oncogene (*GATA2*) exhibited weak DNA–RNA correlation ($r < 0.2$) (Appendix Fig S10A and B). This analysis indicates that gene copy number alterations at the DNA level generally lead to increased RNA expression in signature A tumors in BRCA, LU, and OV tumors, with a similar degree of correlation observed for both glycolysis genes and oncogenes. Finally, we compared the upregulation of glycolytic genes and oncogenes on two cross-species conserved regions of human chr. 12 that contain both types of genes (Appendix Fig S10C). Within the centromere-distal region of chromosome 12, the glycolytic genes *TPI1* and *GAPDH* show strong correlation (Spearman rank correlation = 0.71 and 0.66, respectively) while the oncogene *CHD4* and the glycolytic gene *ENO2* exhibit moderate correlate (Spearman rank correlation = 0.42 and 0.32, respectively). Within the more centromere-proximal region, *LDHB* and *KRAS* both exhibit strong correlation (Spearman rank correlation = 0.68 and 0.66, respectively). Importantly, the correlation of RNA expression with DNA amplification is not stronger for oncogenes than for glycolytic genes within these cross-species regions. Taken together, these results support a model in which the selection pressures shared during immortalization and tumorigenesis result in cross-species conservation of the glycolysis gene loci copy number alterations (Fig 4E and F, and Table EV2).

## Alteration of CNA signatures by exogenous expression of metabolic enzymes

The presence of core glycolysis genes in cross-species conserved amplification regions suggests that these metabolic gene loci drive the amplification of these regions. To test this hypothesis, we transduced pre-senescent MEFs with either wild-type HK2 or HK1, kinase-dead HK2 (D209A/D657A) (McCoy *et al*, 2014), or wild-type ENO2 and allowed the cells to senesce and immortalize in the presence of these exogenously expressed proteins (Appendix Fig S11). Analyzing the signature A set of sublines, we found that the endogenous *Hk2* locus (chr. 6) was less amplified in cells expressing exogenous wild-type hexokinase ($P = 0.048$) (Fig 5A and B). As a control, a signature A MEF cell line expressing kinase-dead hexokinase did not show reduced amplification of the *Hk2* locus ($P = 2 \times 10^{-4}$). In that MEF lines exogenously expressing hexokinase still

demonstrated positive selection for the centromere-proximal half of chr. 6 (Fig 5A), we examined the ratio of *Hk2* gene locus copy number to the maximal amplification on chromosome 6 (Hk2:Chr6 max). In this analysis, we found that cells expressing exogenous hexokinase demonstrated significantly reduced Hk2:Chr6 max ratios ($P = 3 \times 10^{-3}$) (Fig 5A and C). Additionally, a MEF subline expressing exogenous ENO2 exhibited deletion rather than amplification of the *Eno2* locus on chr. 6 ($P = 0.02$) (Fig 5A and D). Analyzing the signature B set of sublines, we found that the endogenous *Hk1* locus (chr. 10) was copy number neutral, rather than amplified, in a cell line expressing exogenous hexokinase ($P = 0.17$), whereas a signature B MEF cell line expressing kinase-dead hexokinase did not show reduced amplification of the *Hk1* locus (Fig 5E and F). Fisher's combined statistical analysis of these results yielded *P*-values of 0.001 or less (Fig 5).

Taken together, these results demonstrate that exogenous expression of metabolic enzymes can alter the copy number status of the endogenous genomic loci, supporting these metabolic genes as drivers within the conserved amplification regions observed in human tumors and mouse models. In addition, these results support a model in which the net propensity for a chromosomal region to be amplified or deleted is in part related to the sum of the fitness effects of the genes present (Davoli *et al*, 2013). For example, when HK2 is exogenously expressed, the centromere-distal half of chromosome 6 had a low copy number value in early culture but after additional culture demonstrated increased copy number, supporting that the other gene loci of this region (such as Eno2, Gapdh, and other glycolytic genes) do have a remaining pro-fitness benefit (Fig 5A).

## Metabolism and growth phenotypes of CNA signatures

To test whether there exist phenotypic differences between signature A and signature B MEFs, we characterized 11 wild-type MEF lines representative of either signature A or B and one mixed signature line. We found that signature A MEFs generally had higher rates of glucose consumption and lactate production than signature B MEF lines (Figs 6A and EV5F). Plotting the PC1 score versus glucose consumption, we found that CNA-based signature A was highly predictive of glucose consumption in the MEF signature A lines (Fig 6B). In contrast, signature B was only moderately predictive of the glucose consumption of signature B lines, and was not as accurate as signature A in predicting the glucose consumption of signature A lines (Figs 6B and EV5A). In addition, we noted that the signature A cell lines generally exhibited significantly higher rates of proliferation than the signature B cell lines (Fig 6C). Similar to glucose consumption, signature A was predictive of the growth rates of signature A MEFs and signature B was predictive of the growth rates of signature B MEFs, while cross-signature predictions had less power (Figs 6D and EV5B). Furthermore, we observed a general coevolution of higher growth rates and increased CNA signature strength in MEF lines that were profiled at different passage numbers (Fig 6D). As noted above, the evolving MEF CNA signature pattern was enriched for DNA amplifications of genes in the core glycolysis and glycolysis-associated pathways (Table EV2), particularly due to amplification of chromosome 6, which contains multiple metabolic gene loci including *Hk2* and *Eno2* (Fig 4A and E, and Appendix Fig S6B).

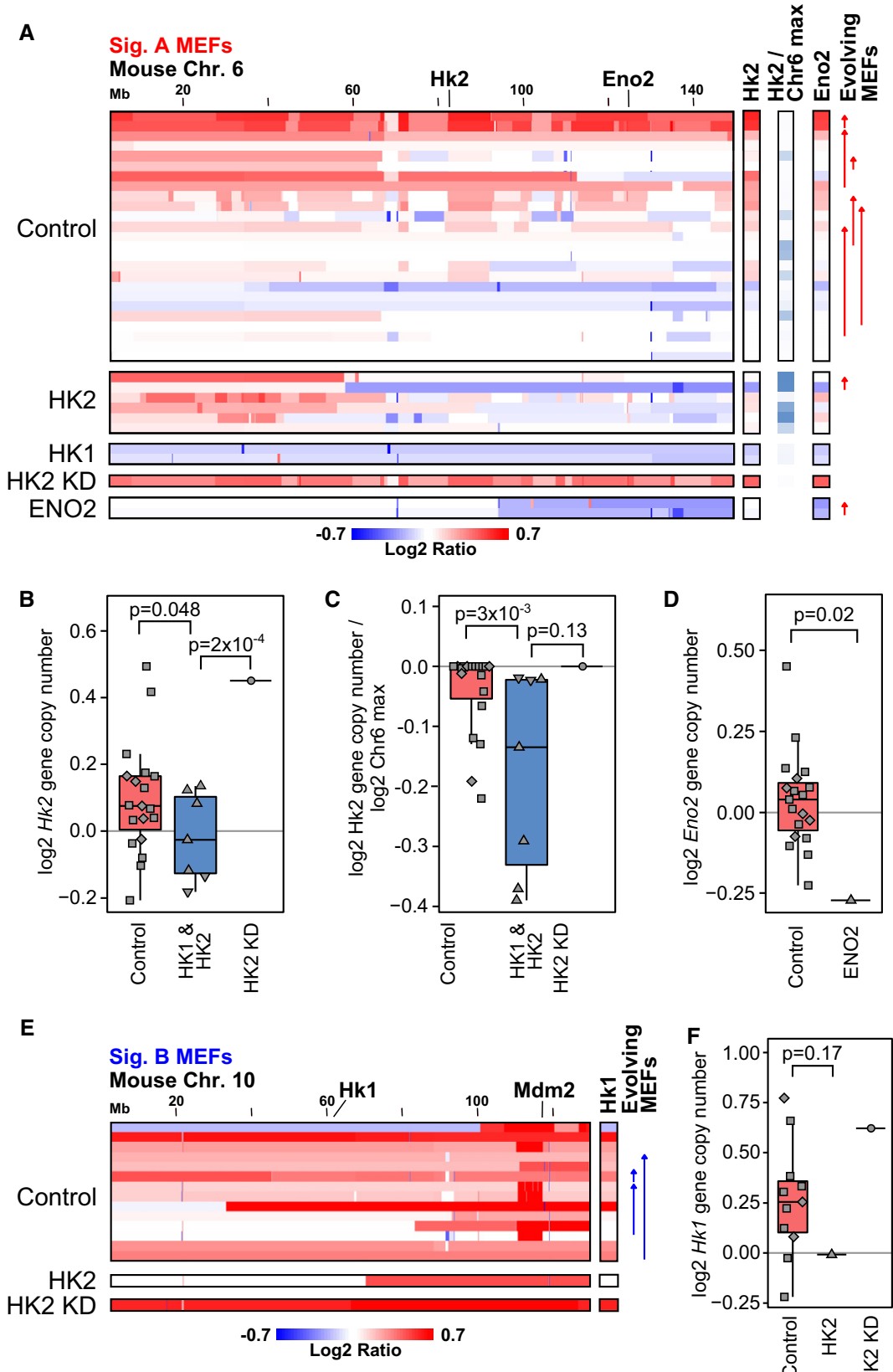

**Figure 5.**

◀

**Figure 5.  Alteration of CNA signatures by exogenous expression of metabolic enzymes.**

A      CD1 MEFs expressing exogenous HK1/HK2 or ENO2 exhibit reduced amplification of the endogenous *Hk2* or *Eno2* loci, respectively. Chr. 6 copy number profiles from control signature A MEFs (untransduced or expressing red fluorescent protein (RFP)) compared to signature A MEFs expressing either wild-type HK1 or HK2, kinase-dead HK2 (HK2 KD, D209A/D657A), or wild-type ENO2. The positions of endogenous *Hk2* and *Eno2* are indicated. MEF lines profiled at more than one passage number are indicated by the start (earlier passage) and end points (later passage) of the arrows under "Evolving MEFs".

B–D    Boxplots of *Hk2* copy number (B), ratio of *Hk2* copy number to maximum amplification of chromosome 6 (defined operationally as the 5$^{th}$ percentile CNA value across chr. 6 to avoid outlier effects) (C), and *Eno2* copy number (D). Control MEFs: untransduced (squares) or expressing RFP (diamonds); test MEFs expressing wild-type HK1 (upside-down triangles), HK2 (triangles in B and C), kinase-dead HK2 (HK2 KD, D209A/D657A, circles), or wild-type ENO2 (triangle in D). In this analysis, copy number data from samples profiled at multiple passages were averaged to prevent overrepresentation of these cell lines. *P*-values were calculated using Student's *t*-test (B), Mann–Whitney *U*-test (C), and *z*-score (B, C) based on criteria described in Materials and Methods. Data are presented in box (median, first and third quartiles) and whisker (extreme value) plots.

E      Signature B MEFs expressing exogenous HK2 exhibit less amplification of the *Hk1* locus. Chr. 10 copy number profiles from control signature B MEFs (untransduced or expressing RFP) compared to signature B MEFs expressing wild-type HK2 or kinase-dead HK2 (HK2 KD). The positions of endogenous *Hk1* and *Mdm2* are indicated.

F      Boxplot of the *Hk1* copy number with indicated *P*-value (*z*-score). Data are presented in box (median, first and third quartiles) and whisker (extreme value) plots.

Data information: Fisher's combined *P*-values for the alteration of endogenous loci CNA by exogenous expression of metabolic enzymes is $3 \times 10^{-5}$ (panels B, D, F) or $7 \times 10^{-4}$ (panels C, D, F). For all comparisons in this figure, no significant differences were observed between untransduced and RFP-expressing MEFs. See also Appendix Fig S11.

To test whether signature A and signature B MEFs differentially use glucose, we cultured MEF cells with [1,2-$^{13}$C]-labeled glucose and conducted metabolomic profiling by mass spectrometry (Fig EV5C–G and Table EV5) (Metallo *et al*, 2009). In all MEF lines tested, we observed a low percentage of single heavy labeled carbon [M1 isotopomer compared to M2] in pyruvate, lactate, and alanine, indicating that the contribution of glucose-derived carbon from the oxidative arm of the pentose phosphate pathway to these metabolites was relatively low (Fig EV5D). Nonetheless, the patterns of heavy isotope labeling revealed differences in nutrient utilization between signature A and B MEF cell lines. On average, metabolites of early glycolysis, the pentose phosphate, and nucleotide synthesis pathways showed a higher percentage of glucose-derived heavy carbon labeling in signature A MEFs (Figs 6E and EV5E). In contrast, signature A cells had a lower percentage of glucose-derived heavy carbon labeling in metabolites of the serine synthesis pathway and the TCA cycle. When compared to signature B MEFs, signature A cells also tended to exhibit increased consumption of serine, glutamine, and other amino acids (Fig EV5F and G). The increased glutamine consumption in signature A MEFs reflects similarity with basal breast cancer cell lines (also signature A associated), which exhibit increased glutamine consumption relative to luminal breast cancer cell lines (Timmerman *et al*, 2013). In general, the percentage of individual metabolites from the early glycolysis, the pentose phosphate, and nucleotide synthesis pathways that incorporated heavy, glucose-derived carbon metabolites was positively correlated with glucose consumption rates (e.g., fructose-1,6-bisphosphate, Fig 6F). Conversely, metabolites from the serine synthesis pathway and the TCA cycle showed a negative correlation between glucose consumption rates and the percentage of each molecule containing a heavy, glucose-derived carbon (e.g., 3-phosphoserine, Fig 6F). While caution must be exercised when interpreting metabolite labeling results, we overall observed a consistent trend of differences between signature A and B lines. Of note, the differences between signature A and B lines were mainly in regard to scale, with the strongest signature B lines demonstrating similar glycolysis and proliferation rates, as well as similar metabolic profiles, as the weakest signature A lines.

Taken together, these results demonstrate that signature A MEFs, which resemble signature A human tumors, exhibit increased glycolysis and to a somewhat lesser extent increased proliferation, and have an increased relative proportion of glucose-derived carbon in metabolites of pentose phosphate-associated biosynthetic pathways such as nucleotide synthesis. These findings are consistent with published mouse model studies in which tumor cells that are channeling glucose toward nucleotide biogenesis achieve faster rates of proliferation (Boros *et al*, 1998; Ying *et al*, 2012).

Thus, our cross-species and pan-cancer CNA analysis revealed conserved amplification regions shared by the majority of tumor types studied that are enriched for genes involved in core glycolysis. To aid others in pan-cancer and cross-species CNA signature comparisons, we have created an interactive web-interface resource available at http://systems.crump.ucla.edu/cna_conservation/.

# Discussion

Chromosomal instability and high glycolysis characterize some of the most aggressive tumors, and the complexity and plasticity of the genomes in aggressive tumors can hinder molecularly targeted therapies (Nakamura *et al*, 2011; McGranahan *et al*, 2012; Shi *et al*, 2012). While the glycolytic changes associated with tumorigenesis were one of the early defining phenotypes of cancer cells (Warburg, 1956), they have not previously been linked to recurrent DNA copy number patterns. Taken together, our experimental and computational data support a model in which glycolysis-linked selective pressures encountered during tumorigenesis (e.g., redox stress and senescence) shape the highly recurrent DNA copy number alterations found in aneuploid human tumors (Fig 6G). We found that CNAs in core glycolysis enzymes (e.g., HK2) and other cancer-linked metabolic enzymes such as TIGAR are coordinately enriched in tumors with distinct CNA signatures. These CNA signatures are predictive of glycolysis, including patient FDG-PET activity and cell line proliferation phenotypes. The strong correlation of CNA-based principle component scores to uptake of the glucose analogue FDG in breast cancer patients and the predictive power of CNA signatures for breast cancer cell metabolism (Fig 3) provide support that the CNAs affecting metabolic gene loci collectively act as a copy number-based driver of metabolic differences. Importantly, in that exogenous expression of hexokinase and enolase enzymes reduced

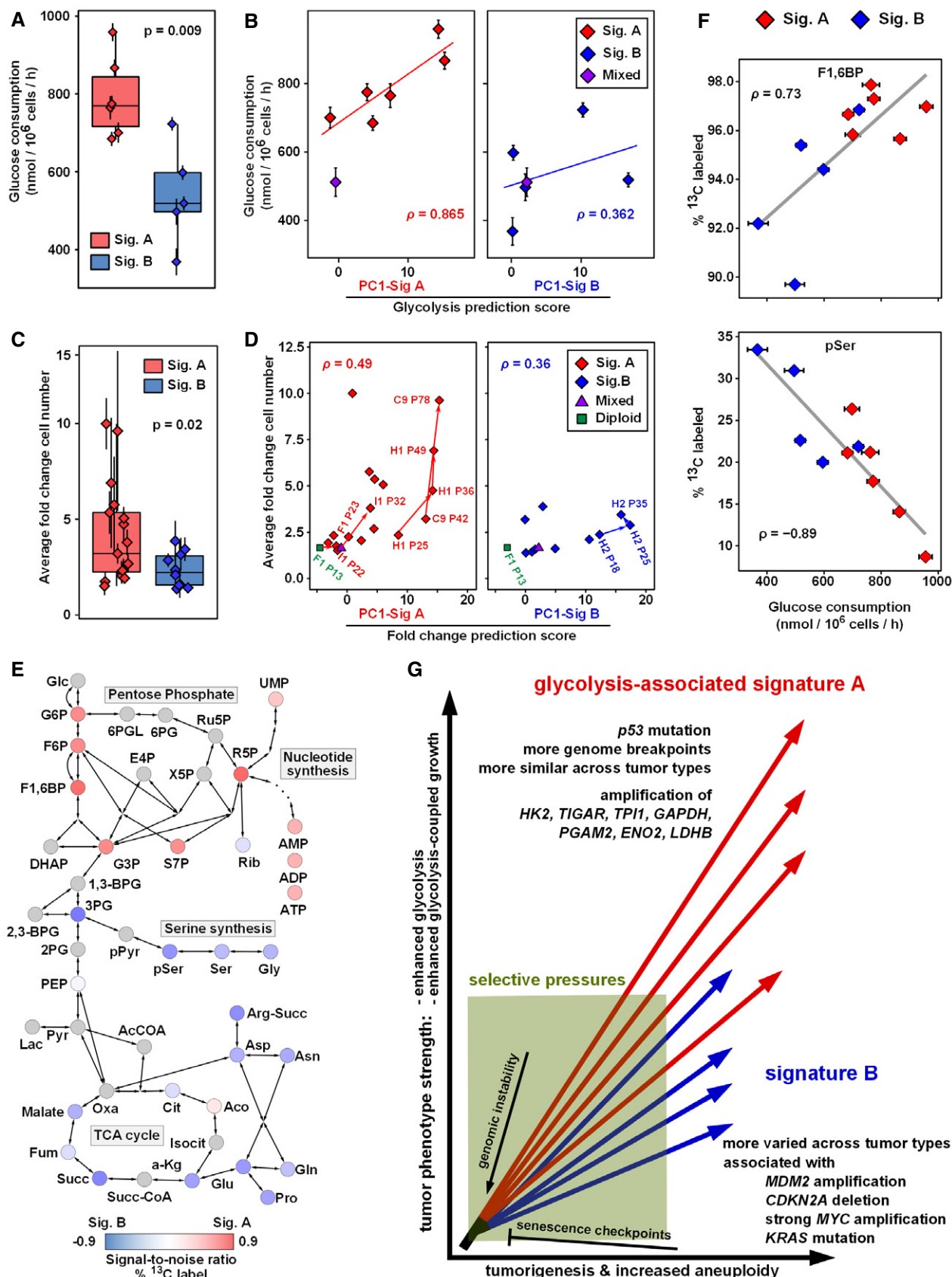

**Figure 6.**

**Figure 6.   Signature A and B MEFs exhibit differential metabolic and proliferation phenotype strengths.**

A   PCA-defined signature A MEFs exhibit higher glucose consumption rates than signature B MEFs (Student's *t*-test *P*-value = 0.009). Glucose consumption was measured using a bioanalyzer.

B   PC1 scores of signature A MEFs are predictive of glucose consumption rates in signature A MEFs, while scores from signature B MEFs have weaker predictive power in signature B MEFs. In cross-prediction tests, signature A-based predictions of signature A MEF line metabolic phenotypes perform the best (Fig EV5A). Glucose consumption was measured using a bioanalyzer.

C   PCA-defined signature A MEFs exhibit higher average growth fold change, as observed in 3T9 culture, compared to signature B MEFs. Signature A MEFs had a subset of cases with high growth rates (average fold change in cell number per passage > 4) that were not observed in the signature B MEFs (hypergeometric *P*-value = 0.02).

D   Correlation and a general coevolution of higher growth rates and increased CNA signature strength. Arrowed lines indicate progressing MEF lines profiled at more than one passage number (Appendix Figs S6A and B, and S9A). In cross-prediction tests, PC1 scores of signature A MEFs are more predictive of average growth fold change in signature A MEFs, while scores from signature B MEFs are more predictive in signature B MEFs (Fig EV5B).

E   Metabolomic profiling of 11 wild-type MEF lines representative of either signature A or B cultured for 24 h with [1,2-$^{13}$C]-labeled glucose. The metabolic pathway schematic is colored based on differences observed in the percent heavy label for intracellular metabolites (defined as percent of metabolite molecules with isotopomer mass greater than the monoisotopic molecular weight, M0) between signature A and B MEFs using the signal-to-noise (SNR) metric. Red indicates a higher heavy carbon-labeling percentage in signature A MEFs, and blue indicates a higher heavy carbon-labeling percentage in signature B MEFs. Signature A MEF lines incorporate more glucose-derived carbon in metabolites from the early glycolysis steps, the pentose phosphate pathway, and nucleotide synthesis, and a smaller fraction of glucose-derived carbon per molecule in metabolites from the later glycolysis steps, the serine synthesis pathway, and the TCA cycle. Full metabolomic data can be found in Table EV5.

F   Correlation of fructose-1,6-bisphosphate (F1,6BP) and negative correlation of 3-phosphoserine (pSer) with glucose consumption rates. Note, the high to low range for percent metabolite molecules with incorporated label varies for each metabolite, but generally does not extend from 0 to 100%. Signature A and B MEFs are colored red and blue, respectively.

G   Model of copy number selection and fitness gains during tumorigenesis. Genomic instability enables fitness gains in tumor metabolism. In human tumors, cancer cell lines, and an experimental MEF immortalization system, immortalization and tumorigenesis lead to multiple CNA signatures that are predictive of the tumor phenotypes of metabolism and proliferation. Senescence-associated redox stress and other tumorigenesis-related constraints select for stronger CNA signatures. A shared high glycolysis-associated signature A is observed in breast, lung, ovarian, and uterine tumors, additional tumor types, mouse models, and the MEF model system, and is linked with a higher range of glycolysis and proliferation phenotypes. Genetic manipulation of glycolysis enzymes leads to alteration of corresponding CNA signature propensities. Signature A and B genomes reflect two distinct trajectories from diploid to tumor aneuploidy. Signature A tumors are enriched for mutations in *p53* and have smaller sized amplification and deletion genomic regions (i.e., have a higher number of genomic breakpoints), potentially providing increased alternative genome options. Signature A involves amplification of several genes in glycolysis-related pathways (such as *HK2, TIGAR, TPI1, GAPDH, ENO2, PGAM2,* and *LDHB*). Signature B CNA patterns occur in generally less glycolytic and proliferative samples and show more variation across different tumor types. In particular, signature B is enriched for *MDM2* amplification and *CDKN2A* loss, strong *MYC* amplification, and *KRAS* mutation and involves alternate hexokinase isoforms (*HK1, HK3*).

Data information: Data in (A, C) are presented in box (median, first and third quartiles) and whisker (extreme value) plots. Error bars indicate standard deviations of biological replicates in (A, B, C and F). See also Fig EV5.

---

the propensities for amplifications of the corresponding endogenous hexokinase and enolase loci, these metabolic genes empirically score as driver loci. However, we cannot exclude the possibility that the observed metabolic differences are in part due to other cancer-associated regulatory changes such as epigenetic events known to affect metabolism (Sebastián *et al*, 2012). Combined with the observation that metabolic genes can facilitate cellular immortalization (Kondoh *et al*, 2005; Kondoh, 2009; Kaplon *et al*, 2013), our results implicate tumor metabolism as an additional fitness measure linked to how genomic instability can enable tumorigenesis.

### Chromosomal instability and aneuploidy—positive and negative impact on tumor cell fitness

Most solid human tumors exhibit both numerical and structural aneuploidy (Holland & Cleveland, 2012). Paradoxically, chromosomal instability can act either as an oncogene or as a tumor suppressor depending on the context (Weaver *et al*, 2007). Moreover, addition of a single chromosome in MEF cells induces a stress response that impairs proliferation and immortalization (Williams *et al*, 2008). However, numerical aneuploidy can lead to chromosomal instability (Nicholson *et al*, 2015) which results in subchromosomal gains and losses as observed in human tumors, in mouse models of cancer, and in immortalized MEF cells. Thus, aneuploidy can cause an initial fitness loss due to the costs of dealing with non-optimized chromosome numbers, gene copy numbers, and resulting proteomic imbalances. However, the associated state of

chromosomal instability enables further evolution of the genome toward fitness gains through refinement of gene copy number levels. In sum, these fitness gains either outweigh or offset the fitness losses due to aneuploidy. In this context, our data support a model (Fig 6G) in which metabolic selection forces and metabolic gene loci contribute to the recurrent patterns of DNA copy number alteration observed in human tumors.

### Redox stress, biomass accumulation, and associated glycolytic changes in tumorigenesis

Tumorigenesis is a complex, multistage process during which cells must acquire the capability to maintain redox balance while accumulating the macromolecular precursors required for proliferation (DeBerardinis *et al*, 2008; Hanahan & Weinberg, 2011). Numerous stimuli, including RAS mutations, matrix detachment, altered metabolism, and hypoxia, induce the accumulation of intracellular ROS (Lee *et al*, 1999; Schafer *et al*, 2009; Weinberg *et al*, 2010; Anastasiou *et al*, 2011). Because increased ROS levels can trigger replicative senescence and subsequent cell cycle arrest (Lee *et al*, 1999; Takahashi *et al*, 2006), tumors must maintain pools of reduced glutathione using NADPH in part produced via the pentose phosphate pathway. Additionally, increased levels of ROS can divert glycolytic flux into the pentose phosphate pathway through, for example, oxidation and inhibition of the glycolytic enzyme PKM2, thereby supplying cells with the reducing power and precursors for anabolic processes (Boros *et al*, 1998; Anastasiou *et al*, 2011).

Consistent with this published knowledge on the role of metabolism in tumorigenesis, our study suggests that the metabolic stress associated with senescence (Fig 4A and D) and the metabolic demands of rapid proliferation (Fig 6) are components of the selective pressures underlying recurrent CNA changes.

## Experimentally and computationally deciphering CNA patterns

Our experimental and bioinformatic approaches complement existing approaches for testing hypotheses for the selection pressures underlying recurrent CNA patterns observed in human tumors. Other CNA-analysis approaches target different resolutions of the genome. Statistical algorithms such as GISTIC (Genomic Identification of Significant Targets in Cancer) have identified many strong individual driver genes and candidate regions (Mermel *et al*, 2011). Integrating CNA data with RNA knockdown screens and gene expression data has further identified driver genes missed by statistical analysis of CNA data alone (Sanchez-Garcia *et al*, 2014). RNA knockdown-based analyses have also been used to support a more systems-level model in which the selection for amplification and deletion of a particular DNA region is based on the cumulative effects of many positive and negative fitness gains from multiple genes within that genomic region (Solimini *et al*, 2012; Cai *et al*, 2016). Subsequent computational extensions have incorporated somatic mutation patterns to infer the cumulative impact of co-localized genes on fitness, and to successfully predict whole chromosome and chromosome-arm resolution-level CNAs (Davoli *et al*, 2013).

Our approach using phenotypic data, functional gene sets, and cross-species syntenic mapping has yielded additional insight into the selective pressures shaping tumor CNA patterns, namely coordinated alteration of genes involved in glycolytic metabolism. Incorporation of copy number data from genetically engineered mouse models allowed synteny constraints to substantially reduce passenger amplifications and define a minimal collection of stringently conserved copy number regions. Additionally, to our knowledge, our experimental approach is the first reported system in which CNA and associated phenotypes have been followed and repeatedly sampled as non-immortalized cells undergo spontaneous genomic instability and proceed from a diploid state to an immortalized aneuploid state. Using this approach, we were able to validate genomic regions that are (i) associated with increased glycolysis and to a lesser but significant extent proliferation, (ii) enriched for genes from the core glycolysis pathway, and (iii) conserved in both human tumors and mouse models of cancer. Metabolic pathways are known to be coordinately regulated by modest changes in mRNA expression of functionally related genes (Mootha *et al*, 2003; Palaskas *et al*, 2011). The coordinated alterations of metabolic genes at the DNA level adds an additional mechanism, namely conserved sets of CNA changes, by which glycolysis is dysregulated to promote tumorigenesis.

A strength of our PCA-based approach is the ability to unbiasedly reveal distinct CNA subsignatures within a tumor type. Observed subsignatures, confirmed using an independent clustering analysis, were found to be associated with previously known pathology- or profiling-defined tumor subtypes (e.g., basal/luminal CNA signatures in BRCA) (Bergamaschi *et al*, 2006) (Figs 1B and EV2A–D). Across multiple tumor types, loss of p53 function through p53

mutation is associated with the high breakpoint signature A pattern. In contrast, loss of the p53/ARF axis via other mechanisms (*MDM2* amplification or *CDKN2A* deletion) in BRCA human tumors is associated with a different CNA signature (signature B) that in general has fewer breakpoints. In our experimental follow-up, the MEF system recapitulated a two-signature pattern. Notably, the two mouse signatures and their associated phenotype strengths were defined by the initiating loss of tumor suppression event, namely *Trp53* mutation versus either *Mdm2* amplification or *Cdkn2a* loss. Thus, while the consequences of *TP53* mutation and "*MDM2* amplification/*CDKN2A* loss" are considered functionally similar and therefore mutually exclusive (Wade *et al*, 2013), our findings indicate they are not fully equivalent in terms of genomic instability and subsequent metabolic evolution. The tolerance of more highly disrupted and rearranged genomes upon p53 mutation appears to allow more flexibility in the evolution of aneuploid cancer genomes, thereby resulting in stronger glycolysis and somewhat enhanced proliferation. The specific combinations of CNA changes occurring in enzyme isoforms defining a metabolic pathway may be considered "onco-metabolic isoenzyme configurations" with differential potency, and the sets of combinations possible may be limited in part by the degree or specifics of tolerance to genomic rearrangements.

In summary, our work illustrates the value of cross-species comparisons in the analysis of DNA copy number data, much as recent pan-cancer and integrated genomic approaches have uncovered novel cancer subtypes and driver genes (Gatza *et al*, 2014; Hoadley *et al*, 2014). The broken chromosome synteny between human and mouse genomes reduced the size of potential driver regions by fivefold on average, and identified a relatively small number of amplification and deletion regions highly conserved between mouse and human (Figs 2D and 4E). These conserved regions are consistently found across the majority of tumor types examined, have subchromosomal scale with a median size on the order of 10 megabase pairs, and include the glycolysis pathway enzyme CNAs observed in our experimental work.

## Therapeutic and diagnostic implications

The most copy number aberrant tumors tend to have fewer point mutations in canonical oncogenes (e.g., *KRAS*, Fig EV2A and C) and less canonical oncogene amplification (e.g., *MYC*, Fig 1B). Hence, genomic instability and subsequent coordinate alterations in multiple genes within a functional pathway may provide an alternate, more complex, pathway to acquisition of aggressive tumor phenotypes—with tumor evolution and selection guiding the trajectory (Ciriello *et al*, 2013). In that *KRAS* mutation and *MYC* amplification can drive glycolysis (Ying *et al*, 2012; Dang, 2013), the findings that signature A tumors are de-enriched in these events relative to signature B tumors and enriched for glycolysis gene loci CNA amplifications support that tumor cells can meet their metabolic demands through distinct mechanisms, or combinations thereof. Future models of the most aggressive cases of cancer will need to incorporate aspects of spontaneous genomic instability (mediated by distinct instability mechanisms) and resulting copy number alterations. Translationally, understanding how CNA patterns alter cancer genomes and impact cancer phenotypes will aid in the identification of metabolic or other "hard-wired" vulnerabilities that can

be therapeutically targeted. Furthermore, the relative stability of DNA samples, combined with the growing linkage between highly recurrent copy number changes and phenotypes, supports the potential for molecular classification and diagnostic tests based on DNA copy number patterns (Hieronymus *et al*, 2014; Cai *et al*, 2016).

# Materials and Methods

## Cell culture and mouse strains

### Tissue culture

CD1 mouse embryonic fibroblasts, E14.5, were purchased from Stem Cell Technologies. p53$^{fl/fl}$ MEFs were obtained at day E14.5 from p53$^{fl/fl}$ (FVB.129P2-Trp53$^{tm1Brn}$/Nci) crossed with C57BL/6-129/SV mice. MEF cells were maintained in Dulbecco's modified Eagle's medium without pyruvate and supplemented with 10% FBS and 1% SPF. Cells were lifted and re-plated at a density of $9 \times 10^5$ viable cells/60-mm dish every 3–4 days (i.e., 3T9 protocol; Todaro & Green, 1963). Cells were grown in atmospheric oxygen unless otherwise indicated. Breast cancer cell lines were obtained from the laboratory of Frank McCormick and extensively profiled both genomically and transcriptionally by the laboratory of Joe Gray (Neve *et al*, 2006; Hong *et al*, 2016).

### Genetic engineering

Overexpression of HK1, HK2, kinase-dead HK2 (D209A/D657A), or ENO2 glycolysis enzymes and the MYC oncogene or the control protein RFP in CD1 MEFs was accomplished by transduction of non-immortalized cells with pDS-FB-neo retrovirus, followed by selection in 600 μg/ml G418. Deletion of p53 in p53$^{fl/fl}$ MEFs was induced by infection of non-immortalized cells with either retroviral Cre-GFP or Cre-ERT2 plus treatment with 1 μM 4-OHT.

### ROS protection

To protect cells from reactive oxygen species (ROS), cells were cultured either at physiological oxygen concentrations (3% $O_2$), with 250 U/ml catalase from bovine liver (Sigma-Aldrich), or with both 3% $O_2$ and catalase.

## Genome profiling

### Array CGH profiling

Genomic DNA was harvested using the DNeasy kit (Qiagen). DNA from MEF lines and reference genomic DNA from C57BL/6J mouse tissue were hybridized to Agilent SurePrint G3 Mouse CGH $4 \times 180$ k CGH microarray chips at the UCLA Pathology Clinical Microarray Core. Bioconductor analysis tools were used for data processing: Moving minimum background correction and print-tip loess normalization were performed in snapCGH package (Smith *et al*, 2009); circular binary segmentation (with a minimum of three markers per segment) was performed on smoothed and log2-transformed copy number profiles using DNAcopy package (Seshan & Olshen, 2016); segmented data were converted into a matrix by genes for downstream analyses using *mus musculus 9* RefSeq reference genome from 2011.08.11 in CNTools (Zhang, 2016). Copy number profiles are presented using the Integrative Genomics Viewer (IGV) (Thorvaldsdóttir *et al*, 2013). CNA public datasets and TCGA tumor type abbreviations are available in the Data availability section.

## Bioinformatic and statistical analysis

### PCA

Principal component analysis (PCA) was performed using the mean-centered matrix of CNA values per gene locus. Genes with identical profiles across samples were collapsed to a single representative gene. Techniques such as PCA and the related singular value decomposition (SVD) have been applied to copy number data previously (Sankaranarayanan *et al*, 2015).

### Mutation and tumor subtype enrichment analysis

To test for enrichment of mutations or tumor subtypes within CNA-defined PC scores, TCGA tumors were queried for mutations and tumor subtype designations using the cBioPortal for Cancer Genomics (Gao *et al*, 2013). For each tumor type, we included all genes with significant *q*-values as calculated by MutSigCV (Lawrence *et al*, 2013). We also included the gene *POLE*, a catalytic subunit of DNA polymerase epsilon, in our analysis because of its frequent mutation in UCEC (The Cancer Genome Atlas Research Network, 2013). Molecular subtypes tested were as follows: BRCA: basal, luminal, claudin-low, and HER2-enriched (The Cancer Genome Atlas Research Network, 2012b); LUAD: bronchioid, magnoid, and squamoid (The Cancer Genome Atlas Research Network, 2014); LUSC: basal, classical, primitive, and secretory (The Cancer Genome Atlas Research Network, 2012a); OV: proliferative, immunoreactive, differentiated, and mesenchymal (The Cancer Genome Atlas Research Network, 2011); and UCEC: *POLE* ultramutated, microsatellite instability hypermutated, copy number low, and copy number high (The Cancer Genome Atlas Research Network, 2013). Tumors were sorted by their PC1 score, and we calculated a Kolmogorov–Smirnov statistic against the expected distribution of mutations or tumor subtypes. The statistical significance of enrichment was determined by permutation analysis.

### Hierarchical clustering

The TCGA tumors were hierarchically clustered using the pheatmap package in R (Kolde, 2015). Prior to clustering, the CNA values were filtered for gene loci that had zero values across all tumor samples. Non-centered and non-scaled tumor samples were then clustered using centered Pearson correlation distance and Ward's method (with dissimilarities squared before cluster updating). The strength of concordance between hierarchical clustering results and PCA-based signatures was assessed using the hypergeometric *P*-value.

### Consistency signatures

Consistency signatures of conserved amplification and deletions regions were determined using stringent consistency criteria. The signed absolute minimum consistency score (SAMCS) was defined as non-zero when all CNA summary signatures have the same sign across all tumor types, and the score is derived from the absolute value-based minimum summary metric and then re-signed positive for amplification or negative for deletion. Consistent amplifications or deletions were combined into a "consistent region", when absolute SAMCS values greater than 0.05 spanned at least 1 Mbp. Any

two consistent regions separated by < 1 Mbp were combined into a single consistent region.

### Enrichment analysis and weighted gene voting (WGV)

Metabolic pathway enrichment analysis (gene set enrichment analysis, GSEA; Subramanian et al, 2005) and pathway-specific weighted gene voting (WGV) prediction analysis (Golub et al, 1999) were performed using 75 metabolic pathways defined by the Kyoto Encyclopedia of Genes and Genomes (KEGG) database (Kanehisa et al, 2014), using pathways with seven or more measured genes. In the RNA-based enrichment analysis, we included a gene set consisting of genes from the glycolysis and pentose phosphate pathways that were upregulated in FDG-high BRCA tumors, as defined by our previous work (Palaskas et al, 2011). For CNA data, an expanded run of enrichment analysis was performed using 1,321 canonical pathways (CP) of seven or more measured genes as defined by the Broad Institute's Molecular Signatures Database (MSigDB). For KEGG-based metabolic pathway enrichment analysis of CNA data, we collapsed metabolic isoenzyme loci (genes with the same enzyme activity; Enzyme Commission [EC] numbers) that were within 100 kilobases from each other into a single representative locus. For mRNA expression analysis, metabolic isoenzymes were not collapsed. In CNA-based enrichment analysis using gene-based versions of the genome consistency signatures defined above, enrichment scores were calculated through the ranked set of consistently amplified genes since after this point the genes that are not consistent across signatures have a consistency value of zero and accordingly have tied ranks. Consistency regions were stepwise restricted by sequentially adding human, and corresponding tissue-matched mouse model of cancer, signatures based on their decreasing tissue type PC1 value from Fig 1A. To combine BRCA and LU into one enrichment analysis for mRNA data, genes were ordered by their average rank in BRCA and LU tumor types. For WGV predictions of metabolic phenotypes, t-scores were used as gene weights. To calculate permutation P-values, we calculated the fraction of 1,000 randomly chosen gene sets of equal size that gave average gene set rankings (column 2 of Fig 3C) better than the true gene set. False discovery rate (FDR) q-values were calculated using the Benjamini–Hochberg procedure. To increase statistical stringency, FDR values for each individual glycolysis–gluconeogenesis gene set (KEGG Glycolysis-Gluconeogenesis, KEGG Core Glycolysis, KEGG Glycolysis-Gluconeogenesis & Pentose Phosphate Pathway, Reactome Gluconeogenesis, Reactome Glucose Metabolism, Reactome Glycolysis, Biocarta Glycolysis Pathway) were calculated while removing the other glycolysis–gluconeogenesis gene sets. CNA and metabolite changes were visualized in the context of metabolic pathway structure using Cytoscape (Smoot et al, 2011).

### Core glycolysis and glycolysis-associated gene list (from Fig 1F)

HK—hexokinase (HK1, HK2, HK3, HK4/glucokinase/GCK); GPI—glucose-6-phosphate isomerase; G6PC—glucose-6-phosphatase catalytic subunit (G6PC, G6PC2); FBP—fructose-bisphosphatase (FBP1, FBP2); PFK—phosphofructokinase (PFKL—liver type, PFKM—muscle, PFKP—platelet); PFKFB—6-phosphofructo-2-kinase/fructose-2,6-biphosphatase (PFKFB1, PFKFB2, PFKFB3, PFKFB4); TIGAR—TP53 induced glycolysis regulatory phosphatase; ALDO—aldolase, fructose-bisphosphate (ALDOA, ALDOB, ALDOC); TPI1—triosephosphate isomerase 1; GAPDH—glyceraldehyde-3-phosphate dehydrogenase (GAPDH, GAPDHS—spermatogenic); PGK—phosphoglycerate kinase (PGK1, PGK2); PGAM—phosphoglycerate mutase (PGAM1, PGAM2, PGAM4, BPGM—bisphosphoglycerate mutase); ENO—enolase (ENO1, ENO2, ENO3); PK—pyruvate kinase (PKLR—liver and red blood cell, PKM2—muscle); LDH—lactate dehydrogenase (LDHA, LDHB, LDHC, LDHAL6A—LDH A-like 6A, LDHAL6B—LDH A-like 6B); and PDH—pyruvate dehydrogenase (PDHA1, PDHA2, PDHB, DLD—dihydrolipoamide dehydrogenase, DLAT—dihydrolipoamide S-acetyltransferase).

### Metrics

Signal-to-noise ratio (SNR) = $(\mu_1 - \mu_2)/(\sigma_1 + \sigma_2)$, t-score = $(\mu_1 - \mu_2)/\mathrm{sqrt}(\sigma_1^2/n_1 + \sigma_2^2/n_2)$, where $\mu$ = mean, $\sigma$ = standard deviation, $n$ = number of samples.

### Correlation of mRNA and CNA

Copy number alteration and mRNA expression levels for genes found in the cross-species conserved regions in Fig 4E were compared by calculating the Spearman rank correlation coefficient. BRCA, LU, and OV samples with paired mRNA and CNA data were included. UCEC tumors were excluded because there were not sufficient UCEC samples with paired RNA expression data (26% of all samples with CNA data). Known oncogenes were identified by comparison with the Catalogue Of Somatic Mutations In Cancer database (COSMIC, http://cancer.sanger.ac.uk/cosmic; Forbes et al, 2015).

### Senescence score

A summary senescence score for each MEF subline was calculated by subtracting the area under the MEF's growth curve, Z(x), in $\log_2$ scale, from the area under an ideal growth curve, Y(x) (i.e., consistent growth at the fastest observed rate). The area difference was then averaged by dividing by the passage number, $p$, and $\log_2$-transformed, resulting in a normally distributed score.

$$Senescence\,Score = log_2 \frac{\sum_{x=2}^{p}\left(\frac{(Y_{(x)}+Y_{(x-1)})}{2} - \frac{(Z_{(x)}+Z_{(x-1)})}{2}\right)}{p}$$

### Integrated CNA

A genomic instability score termed "integrated CNA" was calculated by summation of the Circular Binary Segmentation algorithm-inferred absolute mean copy number of segments multiplied by the length of each segment.

$$\mathrm{Int.CNA_{sample}} = \frac{\sum_{segments} |\text{segment end} - \text{segment start}| \times |\text{segment mean}|}{\#\text{base pairs in sample}}$$

### Box and whisker plots

In box and whisker plots, the box represents the median, as well as the first and third quartiles, and the whisker indicates the extreme values within 1.5 times the inter-quartile range. In cases where the number of samples permitted, individual values are superimposed as jitter plots.

### Statistical tests

Indicated P-values were calculated using (i) Student's t-test for normally distributed data (with normality confirmed by the P-value

of the Shapiro–Wilk test being > 0.05 for both datasets under comparison); (ii) Mann–Whitney *U*-test for non-normally distributed data; (iii) hypergeometric distribution *P*-values; and (iv) permutation-based approaches, as described in the figure legends. For data with a single data point in one comparison group, the *z*-score was used.

## Propidium iodide staining

Mouse embryonic fibroblasts cells were washed in cold PBS and then fixed in ice-cold 70% ethanol. Fixed cells were washed in PBS and then incubated for 15 min in PBS with 20 µg/ml propidium iodide and 0.1% (v/v) Triton X-100. Data were acquired using a FACSCalibur (Becton Dickinson) analytic flow cytometer in the UCLA Jonsson Comprehensive Cancer Center and Center for AIDS Research Flow Cytometry Core Facility. Cells were gated using forward scatter and side scatter to remove debris and dead cells, and 10,000 cell events were recorded.

## Spectral karyotyping

Exponentially growing MEF cells were exposed to colcemid (0.04 µg/ml) for 1 h at 37°C and to hypotonic treatment (0.075 M KCl) for 20 min at room temperature. Cells were fixed in a mixture of methanol and acetic acid (3:1 by volume) for 15 min, and then washed three times in the fixative. Slides were prepared by dropping the cell suspension onto wet slides followed by air-drying. Slides were processed for spectral karyotyping (SKY) according to the manufacturer's protocol with slight modifications using mouse paint probes (ASI, Vista, CA). Images were captured using Nikon 80i microscope equipped with spectral karyotyping software from ASI, Vista, CA; 12–18 metaphases were karyotyped from each cell line.

## Immunoblot analysis

Cells were lysed in modified RIPA buffer (50 mM Tris–HCl (pH 7.5), 150 NaCl, 10 mM β-glycerophosphate, 1% NP-40, 0.25% sodium deoxycholate, 10 mM sodium pyrophosphate, 30 mM sodium fluoride, 1 mM EDTA, 1 mM vanadate, 20 µg/ml aprotinin, 20 µg/ml leupeptin, and 1 mM phenylmethylsulfonyl fluoride). Whole-cell lysates were resolved by SDS–PAGE on 4–15% gradient gels and blotted onto nitrocellulose membranes (Bio-Rad). Membranes were blocked overnight and then incubated sequentially with primary and either HRP-conjugated (Pierce) or IRDye-conjugated secondary antibodies (Li-Cor). Blots were imaged using the Odyssey Infrared Imaging System (Li-Cor). Protein levels were quantitated using ImageJ (http://imagej.nih.gov/ij/). Primary antibodies used for Western blot analysis included hexokinase 1 (2024, Cell Signaling Technology), hexokinase 2 (2867, Cell Signaling Technology), p53 (NB200-103, Novus Biologicals), and enolase 2 (8171, Cell Signaling Technology).

## Glucose consumption and lactate secretion measurements

### BRCA cell lines

Lactate secretion rates of breast cancer cell lines were measured from the culture media using a colorimetric assay kit (BioVision) (Hong *et al*, 2016).

### MEFs

Glucose consumption and lactate secretion rates of MEFs were measured using a BioProfile Basic bioanalyzer (NOVA Biomedical). Data were normalized to the integrated cell number, which was calculated based on cell counts at the start and end of the time course and an exponential growth equation. Because the proliferation rates of MEF sublines vary, each cell line was seeded at the appropriate density so as to give an integrated cell number of approximately $6.5 \times 10^5$ cells in a 6-well plate. All samples were run as biological triplicates, and consistent results were seen in multiple independent experiments.

## Mass spectrometry-based metabolomic analyses

### Sample preparation

Mouse embryonic fibroblasts sublines were seeded onto 6-well plates, and after 24 h, media was replaced with media containing 4.5 g/l [1,2-$^{13}$C]-labeled glucose. Sample collection occurred after 24 h of culture in the labeled glucose media. For intracellular metabolite analysis, cells were washed with ice-cold 150 mM ammonium acetate ($NH_4AcO$) pH 7.3 and metabolites extracted in 1 ml ice-cold 80% MeOH. The cells were quickly transferred into a microfuge tube, and 10 nmol norvaline was added to the cell suspension for use as an internal standard. The suspension was subsequently vortexed three times over 15 min and then spun down at 4°C for 5 min. The supernatant was transferred into a glass vial, the cell pellet was re-extracted with 200 µl ice-cold 80% MeOH and spun down and the supernatants were combined. Metabolites were dried at 30°C under vacuum and re-suspended in 50 µl of 70% acetonitrile (ACN). For cell culture media metabolite analysis (footprint profiling), 20 µl of cell-free media samples was collected. Metabolites were extracted by adding 300 µl ice-cold 80% methanol, followed by vortexing three times over 15 min, and centrifugation for 10 min at 13,000 rpm at 4°C. The supernatant was transferred to a fresh tube, dried using a vacuum evaporator, and re-suspended in 50 µl of 70% acetonitrile (ACN); 5 µl was used for mass spectrometry-based analysis.

### Mass spectrometry runs

Samples were run on a Q-Exactive mass spectrometer coupled to an UltiMate 3000RSLC UHPLC system (Thermo Scientific). The mass spectrometer was run in polarity switching mode (+3.00 kV/ −2.25 kV) with an m/z window ranging from 65 to 975. Mobile phase A was 5 mM $NH_4AcO$, pH 9.9, and mobile phase B was ACN. Metabolites were separated on a Luna 3 µm NH2 100 Å (150 × 2.0 mm) (Phenomenex) column. The flow was kept at 200 µl/min, and the gradient was from 15% A to 95% A in 18 min, followed by an isocratic step for 9 min and re-equilibration for 7 min.

### Data analysis

Metabolites were detected and quantified as area under the curve (AUC) based on retention time and accurate mass (≤ 3 ppm) using the TraceFinder 3.1 (Thermo Scientific) software. Relative amounts of metabolites between various conditions, percentage of metabolite isotopomers (relative to all isotopomers of that metabolite), and percentage of labeled metabolite molecules (isotopomer M1 and greater, relative to all isotopomers) were calculated and corrected

for naturally occurring $^{13}$C abundance (Yuan *et al*, 2008). Footprinting data were normalized to the integrated cell number as described above, and intracellular metabolite concentrations were normalized to the number of cells present at the time of extraction. All samples were run as biological triplicates, and consistent results were seen in independent experiments. Our analysis focused on metabolite level, percent isotopomer, and percent labeled metabolite measurements with ANOVA *P*-values across the sample panel of < 0.05 in individual experiments, and Pearson correlation coefficients across all samples and between independent experiments of > 0.5.

**Patient tumor samples and quantitative FDG-PET imaging**

Breast cancer patient samples with imaged FDG uptake within 4 weeks prior to surgery, excluding patients with secondary breast cancers and recurrent disease, were collected surgically and processed as previously described (Palaskas *et al*, 2011). Of eighteen tumors collected in the original study for RNA microarrays (Palaskas *et al*, 2011), ten samples had sufficient remaining frozen tissue for array CGH profiling. None of the patients received systemic therapy or radiation prior to imaging. $^{18}$FDG tumor uptake was quantified as standardized uptake values (SUVs) and showed the expected wide dynamic range (3.8–18.5). There was no significant difference in patient age, tumor volume, and lymph node involvement between the groups of FDG-high and FDG-low breast cancers. Breast cancers with high $^{18}$FDG-PET SUVs frequently lacked expression of the estrogen receptor (ER) and the progesterone receptor (PR), but hormone receptor-negative tumors were also represented among the tumors with the lowest FDG uptake (Palaskas *et al*, 2011). We excluded lobular breast carcinomas, because they have been shown to take up less FDG than ductal carcinomas (Avril *et al*, 2001) We excluded large breast carcinomas (> 5 cm) and breast carcinomas with multifocal FDG uptake because our protocol did not include tissue autoradiography to direct the molecular tissue analysis to areas of distinct radiotracer retention. This study was approved by the Institutional Review Board (IRB) of Memorial Sloan-Kettering Cancer Center, and all participating patients signed the informed consent.

**Data availability**

*CNA dataset*
Copy number profiling data for wild-type and genetically modified MEF samples and FDG-PET-imaged human breast tumors are available through Gene Expression Omnibus (GEO) accession GSE63306 (https://www.ncbi.nlm.nih.gov/geo/query/acc.cgi?acc = GSE63306).

*CNA conservation web resource*
An interactive website for user-defined pan-cancer and cross-species CNA conservation analysis to perform analysis analogous to that in Figs 1C, 2D, and 4E using any combination of tens of available CNA signatures from human tumors and mouse models (and additional signatures as they become available) and/or the inclusion of uploaded CNA signatures (http://systems.crump.ucla.edu/cna_con servation/). Signatures and genome reference files used in the interactive website are additionally available through the Biostudies repository, accession number S-BSST7.

*Metabolomics dataset*
Provided in Table EV5.

*TCGA CNA dataset*
The Cancer Genome Atlas (TCGA) CNA profiles were downloaded from the TCGA portal in September 2012 (https://cancergenome. nih.gov/). Copy number profiles obtained were pre-processed level 3 data based on human genome 19, with copy number variations (CNVs) removed. TCGA tumor type abbreviations and number of samples analyzed: bladder urothelial carcinoma (BLCA, 97 samples), brain lower grade glioma (LGG, 181), breast invasive carcinoma (BRCA, 873), colon adenocarcinoma (COAD, 447), glioblastoma multiforme (GBM, 593), head and neck squamous cell carcinoma (HNSC, 308), kidney renal clear cell carcinoma (KIRC, 539), lung adenocarcinoma (LUAD, 368), lung squamous cell carcinoma (LUSC, 359), ovarian serous cystadenocarcinoma (OV, 584), prostate adenocarcinoma (PRAD, 171), rectum adenocarcinoma (READ, 168), skin cutaneous melanoma (SKCM, 256), stomach adenocarcinoma (STAD, 162), thyroid carcinoma (THCA, 333), and uterine corpus endometrial carcinoma (UCEC, 492).

*TCGA mRNA expression dataset*
The Cancer Genome Atlas mRNA expression data were downloaded from the TCGA portal in May 2014. Gene-based mRNA expression levels were pre-processed, normalized Level 3 RNA Seq V2 RSEM values. TCGA tumor type abbreviations and number of samples analyzed: BRCA (865), LUAD (357), LUSC (358), OV (263).

*Mouse tumor model CNA datasets*
The genetically engineered mouse models with characterized CNA were obtained from public datasets: mammary (breast) tumors (Brca, 57 samples, GSE30710; 62 samples, GSE43997; 44 samples, GSE27101) (Drost *et al*, 2011; Herschkowitz *et al*, 2012); melanoma (Skcm, 30 samples, GSE58265) (Viros *et al*, 2014); glioblastoma/ high-grade astrocytoma (Gbm, 72 samples, GSE22927) (Chow *et al*, 2011); and prostate tumors (Prad, 18 samples GSE35247; 55 samples, GSE61382) (Ding *et al*, 2012; Wanjala *et al*, 2015). Additionally, *in vitro* epithelial murine cell lines modeling human carcinomas were obtained from public datasets: transformed colon cells (Coad, seven samples, GSE70790) and transformed bladder and kidney cells (Blca and Kirc, 6 and 7 samples, GSE45128; Padilla-Nash *et al*, 2013). Abbreviated mouse tumor names match to the corresponding tissue-based human tumor abbreviations from the TCGA datasets. The data were obtained from GEO and segmented using the algorithm described above. Datasets for which no mm9 genome annotation was available on the repository were lifted over to mm9 using UCSC web tools (Rosenbloom *et al*, 2015).

**Expanded View** for this article is available online.

## Acknowledgements

We thank Harvey Herschman, Judith Campisi, Edward Driggers, Michael Phelps, and Steven Bensinger for critical discussion of the project and experimental approaches. We thank Johannes Czernin for providing the FDG-PET image used in the graphical abstract. Flow cytometry was performed in the UCLA Jonsson Comprehensive Cancer Center (JCCC) and Center for AIDS Research Flow Cytometry Core Facility that is supported by National Institutes of Health awards P30 CA016042 and 5P30 AI028697, and by the JCCC, the

UCLA AIDS Institute, the David Geffen School of Medicine at UCLA, the UCLA Chancellor's Office, and the UCLA Vice Chancellor's Office of Research. We acknowledge the Molecular Cytogenetics Facility, Center for Genetics and Genomics, University of Texas, M.D. Anderson Cancer Center, Houston, Texas. N.A.G. is a postdoctoral trainee supported by the UCLA Scholars in Oncologic Molecular Imagining Program (NCI/NIH grant R25T CA098010). A.M. is supported by a UCLA Eugene V. Cota-Robles Fellowship and a UCLA Dissertation Year Fellowship. S.M.L. is supported by the NCI/NIH (P50 CA086438). T.G.G. is supported by the NCI/NIH (P01 CA168585, P50 CA086306, U19 AI067769), an American Cancer Society Research Scholar Award (RSG-12-257-01-TBE), a Melanoma Research Alliance Established Investigator Award (20120279), the Norton Simon Research Foundation, the UCLA Jonsson Cancer Center Foundation, the National Center for Advancing Translational Sciences UCLA CTSI Grant UL1TR000124, the UC Cancer Research Coordinating Committee, a Concern Foundation CONquer CanCER Now Award, and the UCLA Stein/Oppenheimer Endowment.

## Author contributions

NAG, AM, AL, AC, DB, HRC, IKM, and TGG designed the study. NAG, AM, AL, AC, MF, SC, SZ, AD, LB, JtH, DB, and HRC performed experiments. NAG, AM, AL, AC, NGB, JG, NP, NS, DB, and TGG performed bioinformatic analysis. NAG, AM, AL, CH, CN, RQ, HW, MM, HRC, and TGG provided or derived reagents. NAG, AM, AL, DB, HRC, and TGG performed and analyzed metabolomics. AM and ASM performed spectral karyotyping. EP, SML, and IKM generated patient data. NAG, AM, DB, HRC, IKM, and TGG wrote the manuscript. The laboratories of HRC and IKM contributed equally. Experiments were performed at UCLA.

## Conflict of interest

The authors declare that they have no conflict of interest.

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
