## [Review Process File · Molecular Systems Biology]

Recurrent patterns of DNA copy number alterations in tumors reflect metabolic selection pressures

Nicholas Graham, Aspram Minasyan, Anastasia Lomova, Ashley Cass, Nikolas Balanis, Michael Friedman, Shawna Chan, Sophie Zhao, Adrian Delgado, James Go, Lillie Beck, Christian Hurtz, Carina Ng, Rong Qiao, Johanna ten Hoeve, Nicolaos Palaskas, Hong Wu, Markus Müschen, Asha Multani, Elisa Port, Steven Larson, Nikolaus Schultz, Daniel Braas, Heather Christofk, Ingo Mellinghoff and Thomas Graeber

Corresponding author: Thomas Graeber, UCLA

Review timeline:

Submission date:	08 July 2016
Editorial Decision:	30 August 2016
Revision received:	08 November 2016
Editorial Decision:	05 January 2017
Revision received:	12 January 2017
Accepted:	16 January 2017

Editor: Maria Polychronidou

Transaction Report:

1st Editorial Decision

30 August 2016

Thank you again for submitting your work to Molecular Systems Biology. We have now heard back from the two referees who agreed to evaluate your manuscript. The reviewers appreciate that the topic of the study is potentially interesting. However, reviewer #2 raises substantial concerns, which we would ask you to address in a major revision of the study.

Without repeating the points listed below, reviewer #2, who is an expert in genomics and related data analysis, raises significant concerns regarding the data analysis and the conclusiveness of the study. In particular, s/he points out that robust statistical support and additional analyses are required to convincingly support several of the key conclusions (including the presence of clusters A and B). S/he provides constructive suggestions in this regard.

During our 'pre-decision cross-commenting' process, in which the reviewers see all reports, reviewer #1 stated that in his/her review, s/he focused more on the aspects of the manuscript related to metabolism. S/he also mentioned that the first three points of reviewer #2 are likely addressable. Regarding the fourth point of reviewer #2, reviewer #1 mentioned that while a more detailed analysis of expression data and a comparison to expression of oncogenes is interesting and could be included in the manuscript, in his/her opinion the metabolomics data already provide support for increased glycolytic metabolism. Along these lines, we think that the support provided by the metabolomics data needs to be clearly explained in your response to this point (and of course in the revised manuscript).

REFeree REPORTS

Reviewer #1:

The manuscript by Graham et al. identifies specific copy number variation signatures associated with human tumors and compares them to signatures generated in MEFs they transformed. They focus on a signature associated with p53 loss that involves enrichment of various metabolic genes in glycolysis and the PPP. Their data identifies a strikingly similar CNA signature during in vitro and in vivo transformation in MEFs and human tumors, respectively. Using the MEF cell lines of different signatures the authors go on to compare the metabolism of cells from each group and highlight relatively strong correlations in glucose uptake and flux to specific pathways. These results are particularly intriguing since transformation in vitro and in vivo are conducted under very different nutritional environments. I reviewed this paper some time ago for another journal, and all my comments have now been addressed.

Reviewer #2:

The authors have performed somatic copy number alteration (CNA) profiling of human tumors across different tumor types, pursuing a pan-cancer analysis. Their main finding presented in this manuscript is that different cancers exhibit similarity in their copy-number alteration profiles when evaluated at the level of principal component analysis (PCA), and more specifically that recurrent profiles at the level of PCA correlate with measured glycolytic phenotypes (FDG-avidity in the tumor) as well as with increased proliferation. This primary PCA signature is also associated with p53 mutations, and hence may point towards a preponderance towards genetic instability in the tumor. Overall 26 recurrently altered regions were highlighted, including through cross-species analyses, which contain in addition to well-established cancer drivers 8 enzymes of the glycolysis pathway. Correlation of recurrent CNA profiles with glycolysis is seen in human and mouse tumors, and additionally in a mouse embryonic fibroblast system, which is potentially interesting and may point towards a currently unrecognized role of CNAs to mediate glycolytic phenotypes. Unfortunately, as it currently stands, crucial information provided in this manuscript is unclear or missing, though will need to be included by the authors to make this manuscript fully evaluable for publication.

- The authors use a PCA-based approach to compare copy-number profiles, but I could not find any motivation for using a PCA-based approach versus other comparative approaches for assessing copy-number alterations. PCA-based approaches certainly can have the issue that they are opaque with respect to what actually contributes to the separation of clusters, so more information on how this methodology applies here is warranted. Which CNAs/amplicons do PC 1, 2, 3, and 4 actually separate, or do they merely separate copy-number stable from more unstable (p53 mutated) tumors? Currently, the authors discuss PC1 (principal component 1) and PC3 (principal component 3), but in Fig. 1 do not even show PC2 (principal component 2), rendering this rather crucial aspect of the manuscript somewhat opaque and difficult to evaluate (PC2 specifically, should normally explain more variance than PC3 and hence in principle PC2 should be more informative). Furthermore, how strong really is the evidence of distinct clusters (signature A vs. signature B, and BRCA, LUSC, UCEC, OV vs the numerous other TCGA tumor types) in the principal component analysis? The authors should include results from independent clustering approaches to reassure us that the clusters forming the basis for many of their analyses actually exist (i.e. are separable). Can results identified in BRCA, LUSC, UCEC, OV be replicated in other tumor types?

- Support of reported enrichments with p-values and enrichment factors. There are numerous places in the manuscript, where the authors report enrichments in the main text without referring to actual enrichment factors or p-values (e.g. p. 5: "Signature B BRCA tumors, in contrast, were enriched for..."; p. 6 "tumors are statistically larger by 1.3-7 fold..."; p. 6 "the conserved profile of core signature A tumors was significantly enriched for DNA amplifications of core glycolysis pathway genes"; p.9 "We found a strong correlation between the strength of CNA signature A and the measured FDG-PET..."). Descriptive phrases like X is enriched over Y are of little value if not corroborated by P-value and enrichment factor. Furthermore, the authors should refrain making statements that cannot be backed up by such crucial statistics (e.g. p. 10 "...genes from the glycolysis

and pentose phosphate pathway were most predictive..."; p. 14: "p53^{-/-} cells did not undergo senescence and exhibited less strong copy-number alterations than wild-type signature A MEFs" - these statements need to be omitted if no significant p-value can be provided, or alternatively included with the P-value).

- The authors argue that signature A tumors show enrichment for DNA amplifications of core glycolysis pathway genes, but it is not clear how often amplification is clearly driven by a core glycolysis pathway gene rather than a known oncogene driving the amplification. To evaluate this the authors should perform detailed expression analyses for each amplicon individually to evaluate the level of expression alterations gene by gene, to objectively evaluate how often an amplicon is leading to upregulation of bona fide oncogenes versus upregulation of core glycolysis pathway genes. It is just not clear how often core glycolysis pathway genes simply play a bystander ("passenger") role in amplicons versus actually "driving" the gene amplification process. The TCGA resource includes several thousands (at least 7000 or so) cancer samples with donor-matched copy-number and expression data, thus such analysis should be feasible.

- Both human and mouse CNA signatures A included amplification of Hk2, Bpgm, Rpia, Tigar, Eno2, Tpi1, Gapdh, Ldhd, Kras, and Pgk2 (p. 11). What expression levels are observed in tumors carrying the respective amplicon vs. those not harboring the amplicon, for glycolysis-related genes vs oncogenes (Kras etc) and other genes that are commonly copy-number altered in these amplicons? It really will be important to know whether these expression data point to glycolysis-related genes as likely targets. Copy-number amplification is widely understood as a means of tumors to increase the expression level of 'amplicon target genes'. The authors invest a lot of effort to show in various ways that glycolysis-associated genes are often amplified (and hypothesises that these genes may be 'drivers') - but do they also represent the top-ranked enriched pathway when performing expression analyses which would be a crucial characteristic for driver genes? The experimental data linking metabolic gene expression to CNAs, unfortunately, are all somewhat indirect (and thus do not offer many insights into mechanism, for example), hence further substantiation of results presented here through detailed expression analyses will be very important.

Additional comments

- p. 9: "A similar analysis using all nine expanded signature A tumor types demonstrated equivalent results (data not shown)". These data should be shown in the Supplement.

1st Revision - authors' response

08 November 2016

Report continued on next page.

We thank the reviewers for their comments and helpful ideas that have improved the manuscript. Below we detail the additions and changes incorporated that include CNA clustering analysis results and glycolysis-linked RNA expression signatures that further support our findings of distinct CNA signatures shaped by selection for glycolysis enzyme activity. Additions to the main manuscript text are indicated by a red line in the left margin.

Reviewer #2:

The authors have performed somatic copy number alteration (CNA) profiling of human tumors across different tumor types, pursuing a pan-cancer analysis. Their main finding presented in this manuscript is that different cancers exhibit similarity in their copy-number alteration profiles when evaluated at the level of principal component analysis (PCA), and more specifically that recurrent profiles at the level of PCA correlate with measured glycolytic phenotypes (FDG-avidity in the tumor) as well as with increased proliferation. This primary PCA signature is also associated with p53 mutations, and hence may point towards a preponderance towards genetic instability in the tumor. Overall 26 recurrently altered regions were highlighted, including through cross-species analyses, which contain in addition to well-established cancer drivers 8 enzymes of the glycolysis pathway. Correlation of recurrent CNA profiles with glycolysis is seen in human and mouse tumors, and additionally in a mouse embryonic fibroblast system, which is potentially interesting and may point towards a currently unrecognized role of CNAs to mediate glycolytic phenotypes. Unfortunately, as it currently stands, crucial information provided in this manuscript is unclear or missing, though will need to be included by the authors to make this manuscript fully evaluable for publication.

- The authors use a PCA-based approach to compare copy-number profiles, but I could not find any motivation for using a PCA-based approach versus other comparative approaches for assessing copy-number alterations. PCA-based approaches certainly can have the issue that they are opaque with respect to what actually contributes to the separation of clusters, (i) so more information on how this methodology applies here is warranted. (ii) Which CNAs/amplicons do PC 1, 2, 3, and 4 actually separate, or do they merely separate copy-number stable from more instable (p53 mutated) tumors? (iii) Currently, the authors discuss PC1 (principal component 1) and PC3 (principal component 3), but in Fig. 1 do not even show PC2 (principal component 2), rendering this rather crucial aspect of the manuscript somewhat opaque and difficult to evaluate (PC2 specifically, should normally explain more variance than PC3 and hence in principle PC2 should be more informative). (iv) Furthermore, how strong really is the evidence of distinct clusters (signature A vs. signature B, and BRCA, LUSC, UCEC, OV vs the numerous other TCGA tumor types) in the principal component analysis? (v) The authors should include results from independent clustering approaches to reassure us that the clusters forming the basis for many of their analyses actually exist (i.e. are separable). (vi) Can results identified in BRCA, LUSC, UCEC, OV be replicated in other tumor types?

We respond by expanding on the features and appropriateness of PCA analysis, and by describing new clustering analysis that supports the PCA-based findings. These aspects have now been added to the manuscript as described below. We organized our response based on the added numbering of sub-points above (i to vi).

(i) Principal component analysis provides a multi-dimensional summary of the data, with each component being orthogonal. While PCA results are not always fully interpretable, in our pan-cancer PCA analysis each of the principal components 1-3 can be informatively interpreted (results and additions to the manuscript are summarized in point ii below). Additionally, one can examine the gene loci that contribute to the separation of samples by inspecting the gene loci loadings (the contributions of each gene locus to the principal component). This is similar to examining the contributing gene loci in hierarchical clustering results by inspecting the resulting heatmap and dendrograms. We have also added to the manuscript a reference to previous PCA-based analysis of tumor CNA data (Sankaranarayanan, et al. "Tensor GSVD of patient- and platform-matched tumor and normal DNA copy-number profiles uncovers chromosome arm-wide patterns of tumor-exclusive platform-consistent alterations encoding for cell transformation and predicting ovarian cancer survival," *PLoS ONE*, 10:e0121396, 2015. This reference is single value decomposition (SVD) analysis-based. PCA and SVD are highly related, see for example: J. Shlens, "A Tutorial on Principal Component Analysis," *arXiv:1404.1100*, Apr. 2014, <http://arxiv.org/abs/1404.1100>).

(ii & iii) In addition to the pan-cancer PC1 and PC3 sample scores in Fig. 1A, we now provide the pan-cancer PC2 scores in

Appendix Fig. S1A. To allow examination of the genome regions that contribute to the pan-cancer PCA-based separation of samples, we plot the pan-cancer PC1-3 gene loadings in Fig. 1C and Appendix Fig. S1C. As the reviewer anticipated, the pan-cancer PC1 separates diploid from aneuploid tumors (Appendix Fig. S1B; the PC1 loadings are similar to the tumor type-specific signatures A, as can be seen in the revised Fig. 1C); PC2 primarily separates glioblastoma (GBM) from other tumor types (stated in the original submission, but now shown in and Appendix Fig. S1A); and PC3 tends to separate signature A BRCA and LU tumors from signature B BRCA and LU tumors (with signature A and signature B defined as in the original submission based on the tumor-type specific PCA analyses; Fig. 1A and Appendix Fig. S1D). As a notable example of being able to interpret which gene loci contribute to the PCA-based separation of samples, we observe that the GBM-associated PC2 loadings are chromosome 7 high and focally high for the chr. 7 EGFR locus. Additionally, pan-cancer PC2 loadings are low for chr. 10 including the PTEN locus, and focally low near the chr. 9 CDKN2A locus. These events reflect the major CNA events of GBM (Ohgaki et al., *American J. Pathology* 170:1445). For the tumor type-specific PCA analyses, a summary of the CNA patterns underlying the PC1-2 scores is provided in Appendix Fig. S1D.

(iv-v) As the reviewer advised, we have performed hierarchical clustering using profiles from both (a) all 15 tumor types for a pan-cancer analysis and (b) tumor type-specific profiles to individually analyze the four core signature A tumor types (OV, BRCA, UCEC, and LU). Both pan-cancer and tumor type-specific clustering analyses revealed high concordance with the PCA method that was used in our initial approach. *The new clustering results are in Fig. EV3 and Appendix Fig. S3 (refer to downloaded version if these figures are not displaying fully in a browser window).*

- In the *pan-cancer* case of hierarchical clustering, a multi-tumor cluster enriched in PCA-defined signature A tumors was identified (hypergeometric p-value= 1.1×10^{-70} , Fig. EV3). A cluster enriched in the generally less uniform PCA-defined signature B tumors is also observed (p-value = 4.5×10^{-12}).
- In a second clustering-based analysis, a stronger enrichment was seen when only the top and bottom 10% of tumor type-specific PC1 scored tumors were used for analysis (hypergeometric p-value= 1.7×10^{-180} , Appendix Fig. S3A). This result demonstrates how the tumor type-specific PCA analysis applied individually to BRCA, OV, UCEC, LU and other tumors, identifies pan-cancer tumors (those with high tumor-type specific PC1 scores) that share a highly related signature (Signature A).
- In a third clustering-based analysis, *tumor type-specific clustering* of the 4 core signature A tumor types confirmed separation of tumors into clusters that were enriched for either the tumor type-specific PCA-defined signature A tumors or for the signature B tumors. Such dual signature A and signature B enrichment was observed in all four tumor-type cases run individually (hypergeometric p-values ranging from 10^{-18} to 10^{-55} , Appendix Fig. S3B-C).

We thank the reviewer for suggesting this alternate approach that demonstrates results statistically concordant with the initial PCA-based results.

(vi) Yes, the results that we report for BRCA, LUSC, UCEC, and OV tumors can be replicated in other tumor types, both in the PCA analyses and in the hierarchical clustering analyses. These aspects are perhaps most succinctly seen by viewing the new clustering results, in which the BLCA, HNSC, and STAD tumor types have a subset of samples in the multi-tumor type signature A sub-cluster (with other samples found in either the signature B cluster or the diploid cluster, Fig. EV3). In contrast, COAD / READ tumors and GBM tumors have a unique enough pattern that each tumor type forms its own cluster. In addition to these clustering results, summaries for the PCA-based results for all tumor types are in a new Appendix Fig. S1D.

In summary, the additional analyses requested by the reviewer demonstrate that the signature A CNA pattern can be concordantly identified by different approaches, both PCA-based and hierarchical clustering-based. The added clustering approach provides additional insight and succinctly summarizes various aspects of the results.

- Support of reported enrichments with p-values and enrichment factors. There are numerous places in the manuscript, where the authors report enrichments in the main text without referring to actual enrichment factors or p-values (e.g. p. 5: "Signature B BRCA tumors, in contrast, were enriched for..."; p. 6 "tumors are statistically larger by 1.3-7 fold..."; p. 6 "the conserved profile of core signature A tumors was significantly enriched for DNA amplifications of core glycolysis

pathway genes"; p.9 "We found a strong correlation between the strength of CNA signature A and the measured FDG-PET..."). Descriptive phrases like X is enriched over Y are of little value if not corroborated by P-value and enrichment factor. Furthermore, the authors should refrain making statements that cannot be backed up by such crucial statistics (e.g. p. 10 "...genes from the glycolysis and pentose phosphate pathway were most predictive..."; p. 14: "p53^{-/-} cells did not undergo senescence and exhibited less strong copy-number alterations than wild-type signature A MEFs" - these statements need to be omitted if no significant p-value can be provided, or alternatively included with the P-value).

We thank the reviewer for pointing out that some of our reported enrichment factors and p-values were hard to find, and a couple were missing. We originally provided most of the p-values within the figures or figure legends. We have made the following changes:

p. 5: "In BRCA, signature A tumors were enriched for the basal subtype, p53 point mutations, high numbers of genomic breakpoints and thus sub-chromosomal alterations (Fig. 1B,D and Fig. EV2D; $p < 0.001$, $p < 0.001$, $p = 2 \times 10^{-4}$, respectively). Signature B BRCA tumors, in contrast, were enriched for luminal type tumors ($p < 0.001$) and exhibited amplifications of the oncogenes *MYC* and *MDM2* and deletion of the tumor suppressor *CDKN2A* (Fig. 1B)."

- p-values from the original submission's figures added to manuscript text

p. 6 "The per tumor average segment size for signature B BRCA, LU and UCEC tumors are statistically larger by 1.3 to 7 fold (Appendix Fig. S2C-E; $p = 2 \times 10^{-4}$ or less)."

- p-values from the original submission's figures added to manuscript text

p. 6 "the conserved profile of core signature A tumors (ie, OV, BRCA, UCEC, LU) (Fig. 1C) was significantly enriched for DNA amplifications of core glycolysis pathway genes (Fig. 1E-F; Table EV1; $p = 0.024$)."

- p-value from the original submission's figures added to manuscript text

p.9 "We found a strong correlation between the strength of CNA signature A and the measured FDG-PET standardized uptake values (SUV) (Fig. 3A-B; Pearson $\rho = 0.94$, $p = 5 \times 10^{-5}$).

- added the p-value to the figure legend to accompany the Pearson rho value shown in the figure in the original submission
- added the correlation value and the p-value to the manuscript text

p. 10 (now page 11) "...genes from the glycolysis and pentose phosphate pathway were most predictive of these metabolic phenotypes (Fig. 3C; Table EV4; $p = 0.01$).

- p-value from the original submission's figure added to manuscript text

p. 14 (now page 15-16): "p53^{-/-} cells did not undergo senescence (Olive et al, 2004) and exhibited less strong copy number alterations than wild-type signature A MEFs (Appendix Fig. S7B; $p = 9 \times 10^{-4}$).

- p-value from the original submission's figure added to manuscript text

Additionally, we made the following addition of a correlation value and associated p-value:

p. 15: "Upon calculating the degree of senescence encountered by each subline (senescence metric described in Materials & Methods), we found a correlation between senescence and the degree of copy number alterations (Fig. 4D, Spearman $\rho = 0.66$, $p = 3 \times 10^{-8}$; and Appendix Fig. S9A)."

- correlation value and p-value added to manuscript text and figure legend

We identified several places where we had used statistical tests (eg, Student's t-test) appropriate for normally-distributed data sets on data that were not normally distributed (as evaluated by the Shapiro-Wilk test). We have re-calculated these statistical values using tests that are applicable to non-normally distributed data (eg, Mann-Whitney U-test). All prior trends remain supported by the new p-values. This approach is described in the Materials and Methods.

• The authors argue that signature A tumors show enrichment for DNA amplifications of core glycolysis pathway genes, but it is not clear how often amplification is clearly driven by a core glycolysis pathway gene rather than a known

oncogene driving the amplification. To evaluate this the authors should perform detailed expression analyses for each amplicon individually to evaluate the level of expression alterations gene by gene, to objectively evaluate how often an amplicon is leading to upregulation of bona fide oncogenes versus upregulation of core glycolysis pathway genes. It is just not clear how often core glycolysis pathway genes simply play a bystander ("passenger") role in amplicons versus actually "driving" the gene amplification process. The TCGA resource includes several thousand (at least 7000 or so) cancer samples with donor-matched copy-number and expression data, thus such analysis should be feasible.

We agree with the reviewer that analysis of RNA expression may help distinguish between the 'drivers' and 'passengers' that lead to the amplification of each amplicon, and that oncogenes provide an informative reference point for comparison. For this analysis, we first identified known oncogenes in these cross-species conserved regions using the Catalogue Of Somatic Mutations In Cancer (COSMIC, <http://cancer.sanger.ac.uk/cosmic>). This revealed five oncogenes in the 12 conserved regions: *GATA2*, *MYC*, *TRRAP*, *CHD4* and *KRAS*. Our previous analysis had identified 7 glycolysis genes as defined by KEGG in these same regions: *GCK*, *PGAM2*, *ENO2*, *HK2*, *GAPDH*, *LDHB* and *TPI1*. Response Table 1 lists these genes and their chromosomal locations.

Response Table 1: Oncogenes and glycolytic genes in cross-species conserved amplification regions

Chromosome	Chromosome sub-region annotation	Known oncogene	Glycolysis gene
2	2.1	-	HK2
3	3.1	GATA2	-
7	7.1		GCK, PGAM2
7	7.4	TRRAP	
8	8.1	MYC	-
12	12.1	CHD4	ENO2, GAPDH, TPI1
12	12.2	KRAS	LDHB
19	19.1	-	-

We then compared DNA copy number levels to RNA expression data for these glycolytic genes and oncogenes in the core signature A tumor types. We analyzed breast (BRCA), lung (LU) and ovarian (OV) tumors with paired DNA and RNA data. We excluded uterine (UCEC) tumors from our analysis because there were not sufficient UCEC samples with paired RNA expression data (only 26% of the UCEC tumors in our CNA analysis had paired RNA data whereas 99%, 96% and 45% of BRCA, LU and OV tumors had paired RNA expression data, respectively).

To evaluate whether DNA amplification leads to upregulation of gene expression, we calculated the Spearman rank correlation coefficient for every gene in the cross-species conserved regions. Examining the data from BRCA tumors showed that the glycolytic gene *TPI1* and the oncogene *KRAS* both exhibited strong correlation in signature A BRCA tumors ($r = 0.82$ and 0.61 , respectively). Other glycolytic genes including *HK2* and other oncogenes including *MYC* exhibited moderate correlation in signature A BRCA tumors ($r = 0.42$ and 0.35 , respectively). Notably, the correlation coefficients were strongest in signature A tumors for these four genes compared to correlations across signature B tumors or across all tumors (Response Fig. 1).

Response Figure 1. Individual gene examples of correlation between DNA copy number levels and RNA expression levels for glycolytic genes (*TPI1*, *HK2*) and oncogenes (*KRAS*, *MYC*) in BRCA tumors. Signature A and B tumors (as defined by PCA of the CNA) are colored red and blue, respectively. Spearman correlation values for all tumors (black), signature A tumors (red) and signature B tumors (blue) are shown. Also shown in Appendix Fig. S10.

Averaging the correlation coefficient across BRCA, LU and OV tumor types in signature A tumors revealed that three glycolytic genes (*LDHB*, *TPI1*, and *GAPDH*) and one oncogene (*KRAS*) exhibited strong DNA-RNA correlation ($r > 0.66$) (Response Fig. 2). Three glycolytic genes (*HK2*, *ENO2*, *PGAM2*) and three oncogenes (*MYC*, *CHD4*, and *TRRAP*) exhibited moderate correlation ($0.2 < r < 0.5$). Only one glycolytic gene (*GCK*) and one oncogene (*GATA2*) exhibited weak DNA-RNA correlation ($r < 0.2$). This indicates that gene copy number alterations at the DNA level generally lead to increased RNA expression in signature A tumors in BRCA, LU and OV tumors. Importantly, the correlation of RNA expression with DNA amplification is not stronger for oncogenes than for glycolytic genes within these cross-species conserved CNA regions.

Response Fig. 2. DNA copy number levels and RNA expression levels are highly correlated in cross-species conserved regions demonstrating that glycolytic genes are correlated at similar levels as known oncogenes. Waterfall plot of the Spearman rank correlation values for all cross-species conserved genes shown in Figure 4E, bottommost row. Known oncogenes and glycolytic genes are noted with orange or red coloring, respectively. Also shown in Appendix Fig. S10.

Finally, to address the request for a “detailed expression analyses for each amplicon individually”, we examined these correlations on an amplicon-by-amplicon basis. We focused on chromosome 12 because it contains two distinct regions of cross-species consistency containing both oncogenes and glycolytic genes (Response Fig. 3). Within the centromere distal region of chromosome 12, the glycolytic genes *TPI1* and *GAPDH* show strong correlation (Spearman correlation =

0.71 and 0.66, respectively) while the oncogene CHD4 and the glycolytic gene ENO2 exhibit moderate correlation (Spearman correlation = 0.42 and 0.32, respectively). Within the more centromere proximal region, LDHB and KRAS both exhibit strong correlation (Spearman correlation = 0.68 and 0.66, respectively).

Response Figure 3. Correlation between DNA copy number levels and RNA expression levels in two cross-species conserved amplicons (chromosome 12 regions 1 and 2). Also shown in Appendix Fig. S10.

Thus, on the basis of comparing correlation between RNA expression and DNA copy number levels, we conclude that amplification of both known oncogenes and glycolytic genes leads to comparable upregulation of RNA expression for genes within the cross-species conserved CNA regions. This gene expression analysis complements our other computational and experimental results supporting the hypothesis that glycolytic genes in cross-species conserved regions contribute to shaping the recurrent DNA copy number patterns observed in genomically unstable tumors.

These results have been added to the manuscript as Appendix Fig. S10.

- Both human and mouse CNA signatures A included amplification of Hk2, Bpgm, Rpia, Tigar, Eno2, Tpi1, Gapdh, Ldhd, Kras, and Pkg2 (p. 11). What expression levels are observed in tumors carrying the respective amplicon vs. those not harboring the amplicon, for glycolysis-related genes vs oncogenes (Kras etc) and other genes that are commonly copy-number altered in these amplicons? It really will be important to know whether these expression data point to glycolysis-related genes as likely targets. Copy-number amplification is widely understood as a means of tumors to increase the expression level of 'amplicon target genes'. The authors invest a lot of effort to show in various ways that glycolysis-associated genes are often amplified (and hypothesizes that these genes may be 'drivers') - but do they also represent the top-ranked enriched pathway when performing expression analyses which would be a crucial characteristic for driver genes? The experimental data linking metabolic gene expression to CNAs, unfortunately, are all somewhat indirect (and thus do not offer many insights into mechanism, for example), hence further substantiation of results presented here through detailed expression analyses will be very important.

We thank the reviewer for pointing out the importance of understanding how our identified DNA copy number signatures impact patterns of metabolic gene RNA expression. This point also indirectly raises the issue of whether the identified DNA copy number signatures truly result in alterations to the downstream metabolic phenotype. First, as pointed out by Reviewer #1 in the 'pre-decision cross-commenting' process, we would like to highlight that our metabolomics data (Fig. 6 and Fig. EV5) demonstrates that signature A MEF cells do exhibit increased glycolytic metabolism and use a higher proportion of glucose derived carbon in upper glycolysis and the pentose phosphate pathway. Thus, in this model system, the DNA-based signature A pattern is associated with increased and altered downstream metabolic activity.

Second, we would like to point to our previous work, Palaskas et al. (cited in the manuscript; Palaskas, et al., *Cancer Res.* 71: 5164–5174), which demonstrated that FDG-avid breast cancers and FDG-avid cancer cell lines from multiple tissue types express RNA signatures that are enriched for genes in the glycolysis pathway. In this report, glycolysis was indeed the most enriched metabolic pathway in RNA expression analysis. The Palaskas et al. publication also reported our initial observation that FDG-avid tumors have increased DNA copy number alterations.

In order to further address the reviewer's question about whether glycolysis-associated pathways are also enriched in RNA expression data, we compared signature A tumors to signature B tumors using enrichment analysis (GSEA). We analyzed both BRCA and LU tumors because these tumor types show distinct signature A and signature B sub-types. (As described in the manuscript, OV signature B is highly similar to the pan-cancer signature A pattern, and thus does not provide a differential test. UCEC tumors were not included due to a lack of sufficient paired RNA and DNA profiling data.) In the enrichment analysis, we included a gene set consisting of genes from the glycolysis and pentose phosphate pathway that were upregulated in FDG-high BRCA tumors, as defined by our previous work (Palaskas, et al.). In the GSEA analysis, the empirically defined FDG-high gene set ranked number one overall (NES = 2.5, permutation p-value = 0.0002, FDR q-value = 0.007), confirming that signature A tumors have glycolysis RNA expression profiles matching those of FDG-high tumors. Furthermore, BRCA and LU signature A tumors were significantly enriched for genes from the full glycolysis and pentose phosphate pathway (overall rank 4th of 76 pathways, NES = 2.0, permutation p-value = 0.007 and FDR q-value = 0.06), as well as other glycolysis-related pathways (Fig. EV4A and Table EV3). This analysis of RNA expression data is consistent with the enrichment of glycolysis-associated pathways at the DNA copy number level, and point to glycolysis-related pathways as selection targets for upregulation in signature A tumors. The RNA-based enrichment analysis is now shown in Figure EV4A and Table EV3.

To further address the reviewer's question about RNA expression of metabolic genes, we adapted the RNA expression signatures from Palaskas et al. to create a continuous variant of the class prediction algorithm known as weighted gene voting (Golub et al., *Science* 286: 531–537). This approach calculates gene weights based on the RNA expression data from measured FDG high and FDG low tumors and uses these weights to predict FDG uptake in additional samples, here applied to the TCGA human tumor data. This approach provides a quantitative prediction for how glycolytic each tumor is based on RNA expression patterns.

We applied this RNA-based approach to predict the glycolytic phenotype of BRCA, LU and OV tumors with paired DNA copy number and RNA expression data. Again, UCEC was excluded due to a lack of sufficient paired RNA and DNA data. We plotted two DNA-based metrics, integrated CNA (a general measure of genomic instability) and principal component 1 (PC1) from the individual tumor type PCA, and then colored each data point according to its RNA-based predicted glycolytic score (Response Fig. 4A-C, next page). BRCA and LU showed a volcano-plot style relationship between integrated CNA and PC1, demonstrating that PCA separates the two signatures of genomic instability with copy number neutral tumors receiving intermediate PC1 scores.

Coloring each tumor according to the RNA-based WGV score revealed that RNA prediction-based glycolytic tumors were associated with the signature A side of PC1. To assign a statistical p-value to this qualitative observation, we compared the RNA-based WGV glycolysis predictions between the PC1-defined DNA copy number groups of signature A and signature B. As shown in Response Figure 4D, signature A tumors are predicted to be significantly more glycolytic than signature B tumors (p-values of 2.4×10^{-17} and 5.7×10^{-18} for BRCA and LU, respectively). OV tumors showed no significant difference, a result consistent with the observation that both signature A and signature B OV tumors have high pan-cancer PC1 values, and thus OV signature B tumors are in fact more 'signature A-like' than signature B BRCA or signature B LU tumors (Fig. EV2B, Appendix Fig. S1D).

To further confirm that the FDG-uptake predictions are impacted by the signature A versus signature B designation independent of their degree of DNA copy number alterations (integrated CNA scores), we next compared WGV scores for ranges of integrated CNA scores (Response Fig. 4E). This allows analysis of the relationship between PC1 and WGV scores at matched ranges of copy number alterations. As shown in Response Figure 4E, both BRCA and LU exhibit strong correlation between the RNA-based WGV score and the DNA-based PC1 score. A permutation-based p-value revealed that BRCA and LU are significant compared to random permutations at a level of $p < 1 \times 10^{-6}$ each. OV was again not significant (see discussion above).

Based on this analysis, we conclude that signature A tumors, which were identified on the basis of PCA using DNA copy number data, are predicted via RNA expression patterns to be significantly more glycolytic than signature B tumors. This is consistent with our experimental observations that MEF cells with signature A-like copy number alterations are more

glycolytic than signature B MEF cells and use an increased proportion of their glucose-derived carbon in the upper branch pathways of glycolysis. In summary, application of a previously established WGV approach based on RNA expression data from highly glycolytic breast tumors supports that signature A tumors (defined by our glycolytic gene-enriched DNA copy number signature) are more glycolytic than signature B or genomically stable tumors.

Response Fig. 4. Analysis of RNA expression patterns in TCGA tumors. Weighted gene voting analysis shows that signature A tumors exhibit RNA expression signatures consistent with increased glycolytic activity. (A-C) RNA expression signatures of high glycolysis (Palaskas *et al*, 2011) were used to predict the glycolytic phenotype of BRCA (A), LU (B), and OV (C) tumors. The color scale was normalized for each tumor type by the mean FDG score for tumors with integrated CNA score > 0.2 . Non-normalized values are shown in the color legend and in panel D. (D) RNA-based WGV predictions of glycolysis show that BRCA and LU Signature A tumors (top 10% PC1) are predicted to be more glycolytic than the tissue type-matched Signature B tumors (bottom 10% PC1). Mann-Whitney U test p-values are shown. (E) Spearman correlation values between RNA-based WGV glycolysis predictions and CNA-based PC1 values, calculated for different range windows of integrated CNA levels to control for the general increase in glycolysis predictions at increased levels of genomic instability (permutation P-values described in Fig EV4 legend). Also shown in Fig. EV4.

As an additional level of confirmation in-between RNA expression and metabolic activity, our genomically unstable MEF

samples had protein levels of HK2 (or HK1) that correlated with the corresponding DNA locus amplification levels (Appendix Fig. S6E-F).

Additional comments

- p. 9: "A similar analysis using all nine expanded signature A tumor types demonstrated equivalent results (data not shown)". These data should be shown in the Supplement.

These results are now included as Appendix Fig. S4A (Pearson $\rho = 0.92$, $p = 2 \times 10^{-4}$).

2nd Editorial Decision

05 January 2017

Thank you for submitting your revised manuscript to Molecular Systems Biology. We have now heard back from reviewer #2 who was asked to evaluate the study. As you will see below, the reviewer thinks that his/her concerns have been satisfactorily addressed and that the study is now suitable for publication.

Before we formally accept the study for publication, we would like to ask you to address the following editorial issues:

- We noticed that you have made some data (CNA conservation analysis) available at your lab website. To ensure long-term archival, we generally recommend depositing data in databases instead of institute/lab websites. Therefore, we would suggest depositing the data to i.e. BioStudies <<https://www.ebi.ac.uk/biostudies>>. The accession number should be included in the Data Availability section. For further information on Data Deposition please refer to our Author Guidelines <<http://msb.embopress.org/authorguide#datadeposition>>.

REFEREE REPORT

Reviewer #2:

This manuscript has been considerably improved compared to the initially submitted version, and is accompanied by a very helpful point-by-point response. I would support its publication

2nd Revision - authors' response

12 January 2017

We are gratified that both the reviewer and you felt that the reviewer's concerns were satisfactorily addressed and that the study is now suitable for publication. As requested, we have made the following changes to the manuscript:

- We have deposited our data at BioStudies, and indicated this in the Data Availability section (accession # S-BSST7). This data is the input data for our interactive website that provides user-defined specific analyses. Since such analyses cannot be done through a repository, we will keep both options available as now indicated in the manuscript. We have indicated a release date of 08.01.2018, but will change this to the manuscript publication date once it is known.
- We have provided the running title "Tumor CNA reflect metabolic selection" on the title page of both the manuscript and the appendix. This running title is also listed in the header text of these files.
- We have incorporated your suggested changes to the synopsis text. We made an addition to the first bullet point.
- References have been formatted according to the *Molecular Systems Biology* style.
- We have verified that the appropriate labels are present for all figures.

3rd Editorial Decision

16 January 2017

Thank you for sending us your revised manuscript. We are now satisfied with the modifications made and I am pleased to inform you that your paper has been accepted for publication.

Corresponding Author Name: Thomas G. Graeber

Manuscript Number: MSB-16-7159